# Does enforcing fairness mitigate biases caused by subpopulation shift?

**Subha Maity***
Department of Statistics
University of Michigan
smaity@umich.edu

**Debarghya Mukherjee***
Department of Statistics
University of Michigan
mdeb@umich.edu

**Mikhail Yurochkin**
IBM Research
MIT-IBM Watson AI Lab
mikhail.yurochkin@ibm.com

**Yuekai Sun**
Department of Statistics
University of Michigan
yuekai@umich.edu

## Abstract

Many instances of algorithmic bias are caused by subpopulation shifts. For example, ML models often perform worse on demographic groups that are underrepresented in the training data. In this paper, we study whether enforcing algorithmic fairness during training improves the performance of the trained model in the *target domain*. On one hand, we conceive scenarios in which enforcing fairness does not improve performance in the target domain. In fact, it may even harm performance. On the other hand, we derive necessary and sufficient conditions under which enforcing algorithmic fairness leads to the Bayes model in the target domain. We also illustrate the practical implications of our theoretical results in simulations and on real data.

## 1 Introduction

There are many instances of distribution shifts causing performance disparities in machine learning (ML) models. For example, Buolamwini and Gebru [7] report commercial gender classification models are more likely to misclassify dark-skinned people (than light-skinned people). This is (in part) due to the abundance of light-skinned examples in training data. Similarly, pedestrian detection models sometimes have trouble recognizing dark-skinned pedestrians [22]. Another prominent example is the poor performance of image processing models on images from developing countries due to the scarcity of images from such countries in publically available image datasets [20].

Unfortunately, many algorithmic fairness practices were not developed with distribution shifts in mind. For example, the common algorithmic fairness practice of enforcing performance parity on certain demographic groups [1, 11] implicitly assumes performance parity on training data generalize to the target domain, but distribution shifts between the training data and target domain renders this assumption invalid.

In this paper, we consider *subpopulation shifts* as a source of algorithmic biases and study whether the common algorithmic fairness practice of enforcing performance parity on certain demographic groups mitigate the resulting (algorithmic) biases *in the target domain*. Such algorithmic fairness practices are common enough that there are methods [1, 2] and software (*e.g.* TensorFlow Constrained Optimization [10]) devoted to operationalizing them. There are other sources of algorithmic biases

---

*Equal Contribution.

35th Conference on Neural Information Processing Systems (NeurIPS 2021).

(*e.g.* posterior drift [17]), but we focus on algorithmic biases caused by subpopulation shifts in this paper. Our main contributions are:

1. We propose risk profiles as a way of summarizing the performance of ML models on subpopulations. As we shall see, this summary is particularly suitable for studying the performance of risk minimization methods.
2. We show that enforcing performance parity during training may not mitigate performance disparities in the target domain. In fact, it may even harm overall performance.
3. We decompose subpopulation shifts into two parts, a recoverable part orthogonal to the fair constraint and a non-recoverable part, and derive necessary and sufficient conditions on subpopulation shifts under which enforcing performance parity improves performance in the target domain (see Section 4.4).

One of the main takeaways of our study is a purely statistical way of evaluating the notion of algorithmic fairness for subpopulation shift: an effective algorithmic fairness practice should improve overall model performance in the target domain. Our theoretical results characterize when this occurs for risk-based notions of algorithmic fairness.

## 2 Problem setup

We consider a standard classification setup. Let $\mathcal{X} \subset \mathbf{R}^d$ be the feature space, $\mathcal{Y}$ be the set of possible labels, and $\mathcal{A}$ be the set of possible values of the sensitive attribute. In this setup, training and test examples are tuples of the form $(X, A, Y) \in \mathcal{X} \times \mathcal{A} \times \mathcal{Y}$. If the ML task is predicting whether a borrower will default on a loan, then each training/test example corresponds to a loan. The features in $X$ may include the borrower's credit history, income level, and outstanding debts; the label $Y \in \{0, 1\}$ encodes whether the borrower defaulted on the loan; the sensitive attribute may be the borrower's gender or race.

Let $P^*$ and $\widetilde{P}$ be probability distributions on $\mathcal{X} \times \mathcal{A} \times \mathcal{Y}$. We consider $\widetilde{P}$ as the distribution of the training data and $P^*$ as the distribution of data in a hypothetical target domain. For example, $P^*$ may be the distribution of data in the real world, and $\widetilde{P}$ is a biased sample in which certain demographic groups are underrepresented. The difference $P^* - \widetilde{P}$ is the distribution shift. In practice, distribution shifts often arise due to sampling biases during the (training) data collection process, so we call $P^*$ and $\widetilde{P}$ unbiased and biased respectively and refer to $P^* - \widetilde{P}$ as the bias (in the training data). Henceforth $\mathbb{E}^*$ (resp. $\widetilde{\mathbb{E}}$) will denote expectation under $P^*$ (resp. $\tilde{P}$). The set of all hypotheses under consideration is denoted by $\mathcal{H}$ and $\ell : \mathcal{Y} \times \mathcal{Y} \mapsto \mathbf{R}_+$ denotes the loss function under consideration. In Section 3 and 4 we assume the set of sensitive attribute $A$ is discrete. The case with continuous $A$ is relegated to the supplementary document (Appendix 2).

## 3 Benefits and drawbacks of enforcing risk parity

To keep things simple, we start by considering the effects of enforcing **risk parity (RP)**. This notion is closely related to the notion of *demographic parity (DP)*. Recall DP requires the *output* of the ML model $h(X)$ to be independent of the sensitive attribute $A$: $h(X) \perp A$. RP imposes a similar condition on the *risk* of the ML model.

**Definition 3.1** (risk parity)**.** *A model $h$ satisfies risk parity with respect to the distribution $P$ on $\mathcal{X} \times \mathcal{Y} \times \mathcal{A}$ if*

$$\mathbb{E}_P\big[\ell(h(X), Y) \mid A = a\big] = \mathbb{E}_P\big[\ell(h(X), Y) \mid A = a'\big] \text{ for all } a, a' \in \mathcal{A} \text{ and all } h \in \mathcal{H}.$$

RP is widely used in practice to measure algorithmic bias in ML models. For example, the US National Institute of Standards and Technology (NIST) tested facial recognition systems and found that the systems misidentify blacks at rates 5 to 10 times higher than whites [21]. By comparing the error rates of the system on blacks and whites, NIST is implicitly adopting RP as its definition of algorithmic fairness.

It is not hard to see that RP is equivalent to linear constraints on the **risk profile** $(R_{\tilde{P}}(h))$ of an ML model with respect to $\widetilde{P}$:

$$R_P(h) \triangleq \big\{ \mathbb{E}_P\big[\ell(h(X), Y) \mid A = a\big] \big\}_{a \in \mathcal{A}}. \tag{3.1}$$

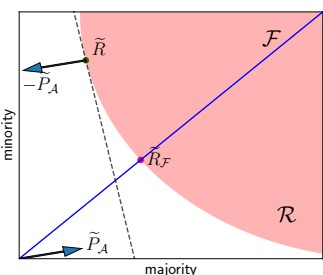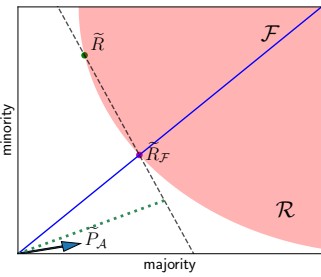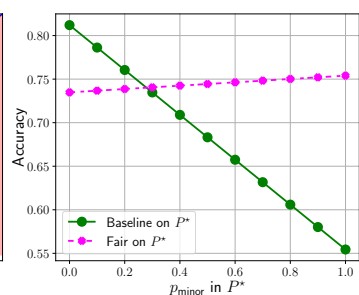

Figure 1: Fair risk minimization problem when there are two groups. Recall $\mathcal{R}$ is a set of risk profiles and $\mathcal{F}$ is the set of risk profiles that satisfy risk parity. The horizontal and vertical coordinates of the risk profiles represent the risk of the model on the majority and minority subpopulations. In the left plot, we see the empirical risk minimization (ERM) optimal point $\widetilde{R}$ and the fair risk minimization (FRM) optimal point $\widetilde{R}_{\mathcal{F}}$. In the center plot, we see that FRM can both improve and harm performance in the target domain (as long as the assumptions of 3.2 are satisfied). The green dotted line separates the $P_A^*$'s that lead to worse and improved performance in the target domain: if $P_A^*$ falls below the green line, then FRM harms performance in the target domain. In the right plots, we reproduce this effect in a simulation. As the fraction of samples from the minority group decreases in the target domain, there is a point beyond which enforcing fairness harms accuracy (in the target domain). We refer to Appendix C for the simulation details.

For notational simplicity define $\mathcal{R}$ as the set of all risk profiles with respect to the training distribution $\tilde{P}$, i.e. $\mathcal{R} \triangleq \{R(h) \mid h \in \mathcal{H}\}$. The risk profile of a model summarizes its performance on subpopulations. In terms of risk profiles, RP with respect to distribution $P$ requires $R_P(h) = c\mathbf{1}$ for some constant $c \in \mathbb{R}$. This is a linear constraint. The set of all risk profiles that satisfy the RP constraint with respect to the training distribution $\tilde{P}$ constitutes the following subspace:

$$\mathcal{F}_{\mathrm{RP}} \equiv \mathcal{F}_{\mathrm{RP}}(\tilde{P}) \triangleq \{R_{\tilde{P}}(h) \in \mathbf{R}^{|\mathcal{A}|} \mid R_{\tilde{P}}(h) = c\mathbf{1}, \mathbf{1} \in \mathbb{R}^{|\mathcal{A}|}, c \in \mathbb{R}\}.$$

Therefore, we enforce RP by solving (the empirical version of) a constraint risk minimization problem

$$\begin{Bmatrix} \min_{h \in \mathcal{H}} & \widetilde{\mathbb{E}}\big[\ell(h(X), Y)\big] \\ \text{subject to} & R_{\tilde{P}}(h) \in \mathcal{F}_{\mathrm{RP}} \end{Bmatrix} \equiv \begin{Bmatrix} \min_{R \in \mathcal{R}} & \langle \widetilde{P}_A, R_{\tilde{P}}(h) \rangle \\ \text{subject to} & R_{\tilde{P}}(h) \in \mathcal{F}_{\mathrm{RP}} \end{Bmatrix}, \tag{3.2}$$

where $\widetilde{P}_A \in [0, 1]^{|\mathcal{A}|}$ is the marginal distribution of $A$ and consequently $\langle \widetilde{P}_A, R \rangle = \widetilde{\mathbb{E}}\big[\ell(h(X), Y)\big]$. Note that, in (3.2), the inner product $\langle \cdot, \cdot \rangle$ is euclidean inner product on $\mathbf{R}^{|\mathcal{A}|}$. We define the minimizer of (3.2) as $\widetilde{h}_{\mathcal{F}}$ and its corresponding risk profile as $\widetilde{\mathcal{R}}_{\mathcal{F}} = R_{\widetilde{P}}(\widetilde{h}_{\mathcal{F}})$. See Figure 1 for a graphical depiction of (3.2) when there are two groups ($\mathcal{A} = \{0, 1\}$). Here we see the main benefit of summarizing model performance with risk profiles: risk minimization problems are equivalent to linear optimization problems in terms of risk profiles (objective and constrain functions are linear in terms of risk profile). This allows us to simplify our study of the effects of enforcing risk parity by reasoning about the risk profiles of the resulting models. Hereafter, we refer to this approach as *fair risk minimization (FRM)*. It is not new; similar constrained optimization problems have appeared in the algorithmic fairness literature (*e.g.* see [1, 12, 10]). Our goal is evaluating whether this approach mitigates algorithmic biases and improves model performance *in the target domain*. There are efficient algorithms for solving (4.3). One popular algorithm is a reductions approach by Agarwal et al. [1], which solves a sequence of weighted classification problems with appropriately chosen weights to satisfy the desired algorithmic fairness constraints. This algorithm outputs randomized classifiers, which justifies the subsequent convexity assumption on the set of risk profiles.

In order to relate model performance in the training and target domains, some restrictions on the distribution shift/bias is necessary, as it is impossible to transfer performance parity during training to the target domain if they are highly disparate. At a high-level, we assume the distribution shift is a *subpopulation shift* [16]. Formally, we assume that the risk profiles of the models with respect to $P^*$ and the profiles with respect to $\tilde{P}$ are identical:

$$\mathbb{E}^*\big[\ell(h(X), Y) \mid A = a\big] = \widetilde{\mathbb{E}}\big[\ell(h(X), Y) \mid A = a\big] \text{ for all } a \in \mathcal{A}, h \in \mathcal{H}. \tag{3.3}$$

i.e. $R_{\tilde{P}}(h) = R_{P^*}(h)$ for all $h \in \mathcal{H}$. We note that this assumption is (slightly) less restrictive than the usual subpopulation shift assumption because it only requires the expected value of the loss (instead of the full conditional distribution of $(X, Y)$ given $A$ to be identical in the training and target domains. Furthermore, this assumption is implicit in enforcing risk-based notions of algorithmic fairness. If the risk profiles are not identical in the training and target domains, then enforcing risk-based notions of algorithmic fairness during training is pointless because performance parity during training may not generalize to the target domain. We are now ready to state our characterization of the benefits and drawbacks of enforcing RP. To keep things simple, we assume there are only two groups: a majority group and a minority group. As we shall see, depending on the marginal distribution of the subpopulations in the target domain $P_A^*$, enforcing RP can harm or improve overall performance in the target domain.

**Theorem 3.2.** *Without loss of generality, let first entry of $\widetilde{P}_A$ be the fraction of samples from the majority group in the training data i.e. $\tilde{P}(A = 1)$. Assume*

1. *there are only two groups and the set of risk profiles $\mathcal{R} \subseteq \mathbf{R}^2$;*
2. *subpopulation shift: the risk profiles with respect to $\widetilde{P}$ and $P^*$ are identical;*
3. *$(\widetilde{R}_1, \widetilde{R}_0) = \widetilde{R} \triangleq \arg\min_{R \in \mathcal{R}} \langle \widetilde{P}_A, R \rangle$ is the risk profile of the risk minimizer;*
4. *$((\widetilde{R}_{\mathcal{F}})_1, (\widetilde{R}_{\mathcal{F}})_0) = \widetilde{R}_{\mathcal{F}} \triangleq \arg\min_{R \in \mathcal{R} \cap \mathcal{F}_{\mathrm{RP}}} \langle \widetilde{P}_A, R \rangle$ is the risk profile of the fair risk minimizer.*

*Then we have:*
$$\langle P_A^*, \widetilde{R} \rangle \begin{cases} \leq \langle P_A^*, \widetilde{R}_{\mathcal{F}} \rangle & \text{if } P^*(A = 1) \geq \frac{\widetilde{R}_0 - (\widetilde{R}_{\mathcal{F}})_0}{\widetilde{R}_0 - \widetilde{R}_1} \\ \geq \langle P_A^*, \widetilde{R}_{\mathcal{F}} \rangle & \text{otherwise}. \end{cases}$$

*Therefore, enforcing RP harms overall performance in the target domain in the first case, while improves in the second.*

In hindsight, this result is intuitive. If $P_A^*$ is close to $\widetilde{P}_A$ (*e.g.* the minority group is underrepresented in the training data but not by much), then enforcing RP may actually harm overall performance in the target domain. This is mainly due to the trade-off between accuracy and fairness in the IID setting (no distribution shift). If there is little difference between the training and target domains, then we expect the trade-off between accuracy and fairness to manifest (albeit to a lesser degree than in IID settings).

## 4 Benefits and drawbacks of enforcing conditional risk parity

### 4.1 Risk-based notions of algorithmic fairness

In this section, we consider more general risk-based notions of algorithmic fairness, namely Conditional Risk Parity (CRP) which is defined as follows:

**Definition 4.1** (Conditional Risk Parity). *a model $h \in \mathcal{H}$ is said to satisfy CRP if:*
$$\mathbb{E}_P\big[\ell(h(X), Y) \mid A = a, V = v\big] = \mathbb{E}_P\big[\ell(h(X), Y) \mid A = a', V = v\big] \tag{4.1}$$

*for all $a, a' \in \mathcal{A}$, $v \in \mathcal{V}$, where $V$ is known as the **discriminative attribute** [18].*

To keep things simple, we assume $V$ is finite-valued, but it is possible to generalize our results to risk-based notions of algorithmic fairness with more general $V$'s (see Appendix B). We also point out that this definition of CRP does not cover calibrations where one conditions on the model outcome $\hat{Y}$.

It is not hard to see that risk parity is a special case of (4.1) in which $V$ is a trivial random variable. Another prominent instance is when $V = Y$, i.e. the risk profile satisfies:
$$\mathbb{E}_P\big[\ell(h(X), Y) \mid A = a, Y = y\big] = \mathbb{E}_P\big[\ell(h(X), Y) \mid A = a', Y = y\big]$$

for all $a, a' \in \mathcal{A}$, $y \in \mathcal{Y}$. Definition 4.1 is motivated by the notion of *equalized odds (EO)* [14] in classification. Recall EO requires the *output* of the ML model $h(X)$ to be independent of the sensitive attribute $A$ *conditioned on the label*: $h(X) \perp A \mid Y$. CRP imposes a similar condition on the *risk* of the ML model; *i.e.* the risk of the ML model must be independent of the sensitive attribute conditioned on the discriminative attribute (with label as a special case). Therefore CRP can

be viewed as a general version of EO, where we relax the conditional independence of $h$ to equality of conditional means. CRP is also closely related to *error rate balance* [9] and *overall accuracy equality* [5] in classification.

Like RP, (4.1) is equivalent to linear constraints on the risk profiles of ML models. Here (with a slight abuse of notation) we define the risk profile of a classifier $h$ under distribution $P$ as:

$$R_P(h) \triangleq \left\{ \mathbb{E}_P\big[\ell(h(X), Y) \mid A = a, V = v\big] \right\}_{a \in \mathcal{A}, v \in \mathcal{V}} \qquad (4.2)$$

Compared to (3.1), (4.2) offers a more detailed summary of the performance of ML models on subpopulations that not only share a common value of the sensitive attribute $A$, but also a common value of the discriminative attribute $V$. The general fairness constraint (4.1) on the training distribution $\tilde{P}$ is equivalent to $R_{\tilde{P}}(h) \in \mathcal{F}_{\mathrm{CRP}}$, where $\mathcal{F}_{\mathrm{CRP}}$ is a linear subspace defined as:

$$\mathcal{F}_{\mathrm{CRP}} \triangleq \{ R_{\tilde{P}}(h) \in \mathbf{R}^{|\mathcal{A}| \times |\mathcal{Y}|} \mid R_{\tilde{P}}(h) = \mathbf{1}\mathbf{u}^\top, \mathbf{1} \in \mathbb{R}^{|\mathcal{A}|}, \mathbf{u} \in \mathbb{R}^{|\mathcal{Y}|}, h \in \mathcal{H} \}.$$

In this section, we study a version of (3.2) with this general notions of algorithmic fairness:

$$\left\{ \begin{array}{ll} \min_{h \in \mathcal{H}} & \widetilde{\mathbb{E}}\big[\ell(h(X), Y)\big] \\ \text{subject to} & R_{\tilde{P}}(h) \in \mathcal{F}_{\mathrm{CRP}} \end{array} \right\} = \left\{ \begin{array}{ll} \min_{R \in \mathcal{R}} & \langle \widetilde{P}_{A,V}, R \rangle \\ \text{subject to} & R \in \mathcal{F}_{\mathrm{CRP}} \end{array} \right\}, \qquad (4.3)$$

where $\widetilde{P}_{A,V} \in [0,1]^{|\mathcal{A}| \times |\mathcal{V}|}$ is the marginal distribution of $(A, V)$, i.e. $\langle \widetilde{P}_{A,V}, R \rangle = \widetilde{\mathbb{E}}\big[\ell(h(X), Y)\big]$. As before we define the minimizer of (4.3) as $\widetilde{h}_{\mathcal{F}}$ and its corresponding risk profile as $\widetilde{\mathcal{R}}_{\mathcal{F}} = R_{\widetilde{P}}(\widetilde{h}_{\mathcal{F}})$. We note that (4.2) has the same benefit as the definition in (3.1): the fair risk minimization problem in (4.3) is equivalent to a linear optimization problem in terms of the risk problems. This considerably simplifies our study of the efficacy of enforcing risk-based notions of algorithmic fairness.

## 4.2 Subpopulation shift in the training data

Similar to equation (3.3), we assume that the risk profiles with respect to $P^*$ and $\widetilde{P}$ are identical:

$$\mathbb{E}^*\big[\ell(h(X), Y) \mid A = a, V = v\big] = \widetilde{\mathbb{E}}\big[\ell(h(X), Y) \mid A = a, V = v\big] \ \ \forall\, a \in \mathcal{A}, v \in \mathcal{V}, h \in \mathcal{H}\,. \qquad (4.4)$$

i.e. $R_{\tilde{P}}(h) = R_{P^*}(h)$ for all $h \in \mathcal{H}$. This definition of subpopulation shift (equation (4.4)) is borrowed from the domain adaptation literature (see [16, 19]). The difference in our definition is that we require equality of the expectations of the loss functions, whereas these works assume the distributions to be equal for the sub-populations. Note that under subpopulation shift $R_{\tilde{P}}$ are $R_{P^\star}$ are equal over $\mathcal{H}$. In the remaining part of Section 4 we shall drop the probability in subscript and denote them as $R$. We note the crucial role of the discriminative attributes in (4.4): the risk profiles are only required to be identical on subpopulations that share a value of the discriminative attribute. A good choice of discriminative attributes keeps the training data informative by ensuring the risk profiles are identical on the training data and at test time. Here are two examples of good discriminative attributes.

**Example 4.2** (Under-representation bias)**.** *In binary classification, training data may suffer from **under-representation bias**. This kind of bias arises when positive examples from disadvantaged groups are under-represented in the training data. Here is an example of a data generating process that suffers from label bias: (i) sample training examples $(X_i, Y_i, A_i)$ from $P^*$, (ii) discard training examples from the disadvantaged group ($A_i = 0$) with positive label ($Y_i = 1$) with probability $\beta$. This leads to*

$$\widetilde{P}(X, Y, A) \propto P^*(X, Y, A) \cdot (1 - (1 - \beta)\mathbf{1}\{A = 0, Y = 1\}).$$

*Because there are fewer positive examples from the disadvantaged group in the training data (compared to test data), this kind of bias causes the ML model to predict mostly negative outcomes for the disadvantaged group. In practice, this kind of bias may creep into the training data more subtly. For example, if human judgements is a crucial part of the data generating process, then implicit biases may lead to over-representation of negative examples from disadvantaged groups in the training data [24].*

*For training data with underrepresentation bias, a good choice of discriminative attribute is the label. This is because the training data is a filtered version of the data at test time, and the filtering process only depends on the label (and sensitive attribute). Thus the class conditionals at test time are preserved in the training data; i.e. $\widetilde{P}_{X|a,y} = P^*_{X|a,y}$ for all $a \in \mathcal{A}$, $y \in \mathcal{Y}$.*

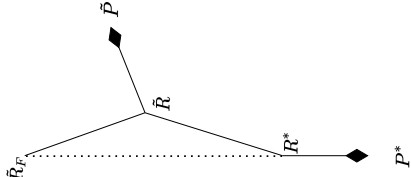

Figure 2: Example in which enforcing algorithmic fairness harms performance in the target domain *despite the Bayes decision rule in the target domain satisfying the algorithmic fairness constraint.*

## 4.3 Fair risk minimization may not improve overall performance

We start by showing that fair risk minimization may not improve overall performance. Without other stipulations, this is implied by a result similar to Theorem 3.2 for more general risk-based notions of algorithmic fairness. Perhaps more surprising, is fair risk minimization may not improve overall performance *even if the Bayes decision rule in the target domain is algorithmically fair*:

$$\arg\min_{R \in \mathcal{R}} \langle P^*_{A,V}, R \rangle \subseteq \mathcal{F}_{\text{CRP}}.$$

Figure 2 shows such a problem instance. The triangle is the set of risk profiles, and the dotted bottom of the triangle intersects the fair constraint (*i.e.* the risk profiles on the dotted line are algorithmically fair). The training objective $\widetilde{P}$ is chosen so that the risk profile of (unconstrained) risk minimizer on biased training data $\widetilde{R}$ is the vertex on the top and the risk profile of fair risk minimizer (also on biased training data) $\widetilde{R}_{\mathcal{F}}$ is the vertex on left. The test objective points to the right, so points close to the right of the triangle have the best overall performance in the target domain. We see that $\widetilde{R}$ is closer to the right of the triangle than $\widetilde{R}_{\mathcal{F}}$, which immediately implies $\langle P^*, \widetilde{R} \rangle \leq \langle P^*, \widetilde{R}_{\mathcal{F}} \rangle$, i.e. it has better performance in the target domain in comparison to $\widetilde{R}_{\mathcal{F}}$. This counterexample is not surprising: the assumption that $R^*$ is fair is a constraint on $P^*$, $\mathcal{R}$, and $\mathcal{F}$; it imposes no constraints on $\widetilde{P}$. By picking $\widetilde{P}$ adversarially, it is possible to have $\langle P^*, \widetilde{R} \rangle \leq \langle P^*, \widetilde{R}_{\mathcal{F}} \rangle$.

## 4.4 When does fair risk minimization improve overall performance

The main result in this section provides necessary and sufficient conditions for recovering the unbiased Bayes' classifier with (4.3). As the unbiased Bayes' classifier is the (unconstrained) optimal classifier in the target domain, enforcing a risk-based notion of algorithmic fairness will improve overall performance in the target domain if it recovers the unbiased Bayes' classifier.

At first blush, it is tempting to think that because the unbiased Bayes classifier satisfies CRP, then enforcing this constraint always increases accuracy, this is not the case as described in the previous paragraph. Our next theorem characterizes the precise condition under which it is possible to improve accuracy on the target domain by enforcing fairness constraint:

**Theorem 4.3.** *Under the assumptions*

1. *The risk set $\mathcal{R}$ is convex.*
2. *The risk profiles with respect to $\widetilde{P}$ and $P^*$ are identical.*
3. *The unconstrained risk minimizer on unbiased data is algorithmically fair; i.e. $\arg\min_{R \in \mathcal{R}} \langle P^*, R \rangle \subseteq \mathcal{F}_{\text{CRP}}$.*

*the fair risk minimization (4.3) obtains $h \in \mathcal{H}$ such that $R(h) = R^*$ if and only if*

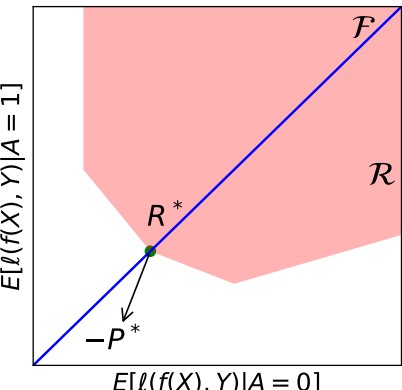

Figure 3: Total recovery from training bias by enforcing risk parity. In this example, the training bias $\widetilde{P} - P^*$ is always orthogonal to the risk parity constraint (blue line) because $\widetilde{P}$ and $P^*$ are probability distributions. When the training bias does not affect the risk profiles, enforcing risk parity allows us to totally overcome the training bias. Unfortunately, to show an example in which the risk decomposes into recoverable and non-recoverable parts, we need (at least) two more dimensions.

$$\Pi_{\mathcal{F}}(P^*_{A,V} - \widetilde{P}_{A,V}) - P^*_{A,V} \in \mathcal{N}_{\mathcal{R}}(R^*) + \mathcal{F}^{\perp}_{\text{CRP}}, \tag{4.5}$$

*where $P_{A,V}^*$ (resp. $\widetilde{P}_{A,V}$) is the marginal of $P^*$ (resp. $\widetilde{P}$) with respect to $(A, V)$, $R^*$ is the optimal risk profile with respect to $P^*$, $\mathcal{N}_{\mathcal{R}}(R^*)$ is the normal cone of $\mathcal{R}$ at $R^*$ and $\Pi_{\mathcal{F}}$ is the projection on the fair hyperplane.*

This assumption that $\mathcal{R}$ is convex is innocuous because it is possible to convexify the risk set by considering *randomized* decision rules. To evaluate a randomized decision rule $H$, we sample a decision rule $h$ from $H$ and evaluate $h$. It is not hard to see that the risk profiles of randomized decision rules are convex combinations of the risk profiles of (non-randomized) decision rules, so including randomized decision rules convexifies the risk set. The third assumption is necessary for recovery of the unbiased Bayes classifier. If the unbiased Bayes classifier is not algorithmically fair, then there is no hope for (4.3) to recover it as there will always be a non-negligible bias term. This assumption is also implicit in large swaths of the algorithmic fairness literature. For example, Buolamwini and Gebru [7] and Yang et al. [23] suggest collecting representative training data to improve the accuracy of computer vision systems on individuals from underrepresented demographic groups. This suggestion implicitly assumes the Bayes classifier on representative training data is algorithmically fair.

Theorem 4.3 characterizes the biases in the training data from which can be completely removed by enforcing appropriate algorithmic fairness constraints. The key insight from this theorem is a decomposition of the training bias into two parts: a part orthogonal to the fair constraint and the remaining part in $\mathcal{N}_{\mathcal{R}}(R^*)$. Enforcing an appropriate risk-based notion of algorithmic fairness overcomes the first part of the bias. This occurs regardless of the magnitude of this part of the bias (see Corollary 4.4), which is also evident from our computational results. The second part of the bias (the part in $\mathcal{N}_{\mathcal{R}}(R^*)$) represents the "natural" robustness of $R^*$ to changes in $P^*$: if $\widetilde{P}$ is in $\mathcal{N}_{\mathcal{R}}(R^*)$, then the unconstrained risk minimizer on training data remains $R^*$. The magnitude of the bias in this set cannot be too large, and enforcing algorithmic fairness constraints does not help overcome this part of the bias. Although we stated our main result only for finite-valued discriminative attributes for simplicity of exposition, please see Appendix 2 for a more general version of Theorem 4.3 that applies to more general (including continuous-valued) discriminative attributes.

**Corollary 4.4.** *A sufficient condition for* (4.5) *is* $\widetilde{P}_{A,V} - P_{A,V}^* \in \mathcal{F}_{CRP}^{\perp}$.

Corollary 4.4 allows large differences between $\widetilde{P}_{A,V}$ and its unbiased counterpart $P_{A,V}^*$, as long as the differences are confined to $\mathcal{F}^{\perp}$. Intuitively, (4.3) enables practitioners to recover from large biases in $\mathcal{F}^{\perp}$ because the algorithmic fairness constraint "soaks up" any component of $\widetilde{P}_{A,V}$ in $\mathcal{F}^{\perp}$. We explore the implications of Corollary 4.4 for risk parity and CRP.

**Risk Parity:** For RP, $V$ is trivial random variable, hence $\widetilde{P}_A - P_A^* \in \mathcal{F}_{RP}^{\perp}$ means that it has mean 0. This is true for any $\widetilde{P}_A$ as $\langle P_A^*, 1 \rangle = \langle \widetilde{P}_A, 1 \rangle = 1$. Hence, the Bayes' classifier can be recovered under any perturbation. More specifically, recall the example of women historically underrepresented in STEM fields mentioned in the Introduction. Such training data is biased in its gender representation which differs at test time where women are better represented. Classifiers trained on biased data with the risk Parity fairness constraint will generalize better at test time.

**Conditional risk parity:** In this case $V = Y$ and the condition $\widetilde{P}_{A,Y} - P_{A,Y}^* \in \mathcal{F}_{CRP}^{\perp}$ implies that the sum of each column of $\widetilde{P}_{A,Y} - P_{A,Y}^*$ must be 0. Hence, to recover the Bayes classifier under equalized odds fairness constraints, we are allowed to perturb $P_{A,Y}^*$ in such a way, that they have the same column sums: i.e. for any label, we are allowed to perturb the distribution of protected attributes for that label, but we have to keep the marginal distribution of the label to be same for both $\widetilde{P}_{A,Y}$ and $P_{A,Y}^*$.

In practice, it is unlikely that the training bias is exactly orthogonal to the fair constraint, which happens only if the second part of the bias (i.e. the part in the normal cone at $R^*$) is small enough. Theorem 4.3 provides a general characterization along with a precise notion of this "small enough" condition.

**Remark 4.5.** *Theorem 4.3 can further be generalized for any discrete/continuous $A$ and $V$ (as defined in (4.1), the proof for the continuous case can be found in the supplementary document). Thus, our theory applies to many fairness constraints which fall under the setup in Equation (4.1), where $V$ can be any discriminative attribute. However, our conditions do not cover calibration where one conditions on the model outcome $\hat{Y}$.*

### 4.5  Related work

Most of the prior works on algorithmic fairness assume fairness is an intrinsically desirable property of an ML model, but this assumption is unrealistic in practice [1, 11, 25]. There is a small but growing line of work on how enforcing fairness helps ML models recover from bias in the training data. Kleinberg and Raghavan [15], Celis et al. [8] consider strategies for correcting biases in hiring processes. They show that correcting the biases not only increases the fraction of successful applicants from the minority group but also boosts the quality of successful applicants. Dutta et al. [13] study the accuracy-fairness trade-off in binary classification in terms of the separation of the classes within the protected groups. They explain the accuracy-fairness trade-off in terms of this separation and propose a way of achieving fairness without compromising separation by collecting more features.

Blum and Stangl [6] study how common group fairness criteria help binary classification models recover from bias in the training data. In particular, they show that the equal opportunity criteria [14] recovers the Bayes classifier despite under-representation and labeling biases in the training data. Our results complement theirs. Instead of comparing the effects of enforcing various fairness criteria on training data with two types of biases, we characterize the types of biases that the fairness criteria help overcome. Our results also reveal the geometric underpinnings of the constants that arise in Blum and Stangl's results. Three other differences between our results and theirs are: (i) they only consider binary classification, while we consider all ML tasks that boil down to risk minimization, (ii) they allow some form of posterior drift (so the risk profiles of the models in $\mathcal{H}$ with respect to $P^*$ and $\widetilde{P}$ may differ in some ways), but only permit marginal drift in the label ($V = Y$), (iii) their conditions are sufficient for recovery of the fair Bayes decision rule (in their setting), while our conditions are also necessary (in our setting).

## 5  Computational results

We verify the theoretical findings of the paper empirically. Our goal is to show that an algorithm trained with fairness constraints on the biased train data $\widetilde{P}$ achieves superior performance on the true data generating $P^*$ at test time in comparison to an algorithm trained without fairness considerations.

There are several algorithms in the literature that offer the functionality of empirical risk minimization subject to various fairness constraints, e.g. Cotter et al. [11] and Agarwal et al. [1]. Any such algorithm will suffice to verify our theory. In our experiments we use Reductions fair classification algorithm [1] with logistic regression as the base classifier. For the fairness constraint we consider Equalized Odds [14] (EO) — one of the major and more nuanced fairness definitions. We refer to Reductions algorithm trained with loose EO violation constraint as baseline and Reductions trained with tight EO violation constraint as fair classifier (please see Appendix 3 for additional details and supplementary material for the code).

**Simulations.**

We first verify the implications of Corollary 4.4 using simulation studies. We follow the Conditional risk parity scenario from Section 4. Specifically, consider a binary classification problem with two protected groups, i.e. $Y \in \{0, 1\}$ and $A \in \{0, 1\}$. We set $P^*$ to have equal representation of protected groups conditioned on the label and biased data $\widetilde{P}$ to have one of the protected groups underrepresented. Specifically, let $p_{ay} = P_{A=a,Y=y}$, i.e. the $a, y$ indexed element of $P_{A,Y}$; $p_{ay} = 0.25 \ \forall a, y$ for $P^*$ and $p_{1y} = p_{minor}, p_{0y} = p_{major} = 0.5 - p_{minor}$ for $\widetilde{P}$. For both $P^*$ and $\widetilde{P}$ we fix class marginals $p_{.0} = p_{.1} = 0.5$ and generate Gaussian features $X|A = a, Y = y \sim \mathcal{N}(\mu_{ay}, \Sigma_{ay})$ in 2-dimensions (see additional data generating details in Appendix C). In Figure 5 we show a

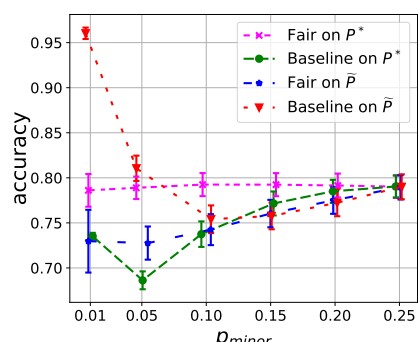

Figure 4: Test acc. on $P^*$ and $\widetilde{P}$ when trained on the (biased) data from $\widetilde{P}$.

qualitative example of simulated train data from $\widetilde{P}$ with $p_{minor} = 0.1$ and test data from $P^*$, and the corresponding decision boundaries of a baseline classifier and a classifier trained with the Equalized Odds fairness constraint (irregularities in the decision heatmaps are due to stochasticity in the Reduc-

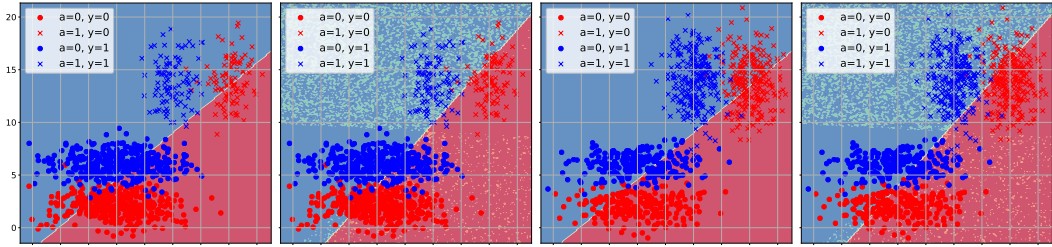

Figure 5: Decision heatmaps for (left) baseline on train data from $\widetilde{P}$; (center left) fair classifier on train data from $\widetilde{P}$; (center right) baseline on test data from $P^*$; (right) fair classifier on test data from $P^*$. Decision boundary of the fair classifier has larger slope better accounting for the group $a = 1$ underrepresented in the train data. Consequently its performance is better on the unbiased test data.

tions prediction rule). In this example fair classifier is *3% more accurate* on the test data and 1% less accurate on a biased test data sampled from $\widetilde{P}$ (latter not shown in the figure)

We proceed with a quantitative study by varying degree of bias in $\widetilde{P}$ via changing $p_{minor}$ in $[0.01, 0.25]$ and comparing performance of the baseline and fair classifier on test data from $P^*$ and $\widetilde{P}$. We present results over 100 runs of the experiment in Figure 4. Notice that the sum of each column of $\widetilde{P}_{A,Y} - P^*_{A,Y}$ is 0 for any value of $p_{minor}$ and we observe that the fair classifier has almost constant accuracy on $P^*$ (consistently outperforming the baseline), as predicted by Corollary 4.4. The largest bias in the training data corresponds to $p_{minor} = 0.01$, where baseline is erroneous on the whole $a = 1, y = 0$ subgroup (cf. Figure 5) resulting in close to 75% accuracy corresponding to the remaining 3 (out of 4) subgroups. For $p_{minor} = 0.05$ minority group acts as outliers causing additional errors at test time resulting in the worst performance overall. When $p_{minor} = 0.25$, $\widetilde{P} = P^*$ and all methods perform the same as expected. Results on $\widetilde{P}$ correspond to the case where test data follows same distribution as train data, often assumed in the literature: here baseline can outperform fair classifier under the more extreme sampling bias conditions, i.e. $p_{minor} \leq 0.1$. We note that as the society moves towards eliminating injustice, we expect test data in practice to be closer to $P^*$ rather then replicating biases of the historical train data $\widetilde{P}$.

**Recidivism prediction on COMPAS data.** We verify that our theoretical findings continue to apply on real data. We train baseline and fair classifier on COMPAS dataset [3]. There are two binary protected attributes, Gender (male and female) and Race (white and non-white), resulting in 4 protected groups $A \in \{0, 1, 2, 3\}$. The task is to predict if a defendant will re-

Table 1: Accuracy on COMPAS data

|  | TEST ON $P^*$ | TEST ON $\widetilde{P}$ |
|---|---|---|
| FAIR | **0.652**±0.013 | 0.660±0.009 |
| BASELINE | 0.634±0.011 | **0.668**±0.010 |

offend, i.e. $Y \in \{0, 1\}$. We repeat the experiment 100 times, each time splitting the data into identically distributed 70-30 train-test split, i.e. $\widetilde{P}$ for train and test, and obtaining test set from $P^*$ by subsampling test data to preserve $Y$ marginals and enforcing equal representation at each of the 4 levels of the protected attribute $A$. Equal representation of the protected groups in $P^*$ is sufficient for satisfying the assumption 3 of Theorem 4.3 under an additional condition that noise levels are similar across protected groups. For Conditional Risk Parity, condition in eq. (4.5) of Theorem 4.3 is satisfied as long as $\widetilde{P}$ and $P^*$ have the same $Y$ marginals. Thus, we expect that enforcing EO will improve test accuracy on $P^*$ in this experiment.

We present results in Table 1. We see that our theory holds in practice: *accuracy of the fair classifier is 1.8% higher* on $P^*$. Baseline is expectedly more accurate on the biased test data from $\widetilde{P}$, but only by 0.8%. We present results for the same experimental setup on the Adult dataset [4] in Table 2 in Appendix C. We observe same pattern: in comparison to the baseline, fair classifier increases accuracy on $P^*$, but is worse on the biased test data from $\widetilde{P}$.

# 6 Summary and discussion

In this paper, we studied the efficacy of enforcing common risk-based notions of algorithmic fairness in a subpopulation shift setting. This study is motivated by a myriad of examples in which algorithmic biases are traced back to subpopulations shifts in the training data (see [7, 22]). We show that enforcing risk-based notions of algorithmic fairness may harm or improve the performance of the trained model in the target domain. Our theoretical results precisely characterize when fair risk minimization harms and improves model performance. Practitioners should be careful and actually check that enforcing fairness is improving model performance in the target domain. For example, consider the Gender Shades [7] study which shows that the commercial gender classification algorithms are less accurate on dark-skinned individuals. A practitioner may attempt to mitigate this algorithmic bias by enforcing a RP, which leads to a fair model that sacrifices some performance on lighter-skinned individuals in exchange for improved accuracy on darker skinned individuals. Taking a step back, one of the main takeaways of our study is by considering whether enforcing a particular notion of algorithmic fairness improves model performance in the target domain, it is possible to compare algorithmic fairness practices in a purely statistical way. We hope this alleviates one of the barriers to broader adoption of algorithmic fairness practices: it is often unclear which fairness definition to enforce.

## Acknowledgments and Disclosure of Funding

This note is based upon work supported by the National Science Foundation (NSF) under grants no. 1916271, 2027737, and 2113373. Any opinions, findings, and conclusions or recommendations expressed in this note are those of the authors and do not necessarily reflect the views of the NSF.

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
