# Appendix of "Does enforcing fairness mitigate biases caused by subpopulation shift?"

**Subha Maity\***
Department of Statistics
University of Michigan
smaity@umich.edu

**Debarghya Mukherjee\***
Department of Statistics
University of Michigan
mdeb@umich.edu

**Mikhail Yurochkin**
IBM Research
MIT-IBM Watson AI Lab
mikhail.yurochkin@ibm.com

**Yuekai Sun**
Department of Statistics
University of Michigan
yuekai@umich.edu

## 1 Supplementary proofs

**Theorem 3.2.** *Without loss of generality, let first entry of $\widetilde{P}_A$ be the fraction of samples from the majority group in the training data. Assume*

1. *there are only two groups and the set of risk profiles $\mathcal{R} \subseteq \mathbf{R}^2$;*
2. *subpopulation shift: the risk profiles with respect to $\widetilde{P}$ and $P^*$ are identical;*
3. *$(\widetilde{R}_1, \widetilde{R}_0) = \widetilde{R} \triangleq \arg\min_{R \in \mathcal{R}} \langle \widetilde{P}_A, R \rangle$ is the risk profile of the risk minimizer;*
4. *$((\widetilde{R}_\mathcal{F})_1, (\widetilde{R}_\mathcal{F})_0) = \widetilde{R}_\mathcal{F} \triangleq \arg\min_{R \in \mathcal{R} \cap \mathcal{F}} \langle \widetilde{P}_A, R \rangle$ is the risk profile of the fair risk minimizer.*

*Then we have:*

$$\langle P_A^*, \widetilde{R} \rangle \begin{cases} \leq \langle P_A^*, \widetilde{R}_\mathcal{F} \rangle & \text{if } P^*(A=1) \geq \frac{\widetilde{R}_0 - (\widetilde{R}_\mathcal{F})_0}{\widetilde{R}_0 - \widetilde{R}_1} \\ \geq \langle P_A^*, \widetilde{R}_\mathcal{F} \rangle & \text{otherwise} . \end{cases}$$

*Therefore, enforcing DP harms overall performance in the target domain in the first case, while improves in the second.*

*Proof of Theorem 3.2.* We start by simplifying $\langle P_A^*, \widetilde{R} - \widetilde{R}_\mathcal{F} \rangle$. Note that

$$\begin{aligned} \langle P_A^*, \widetilde{R} - \widetilde{R}_\mathcal{F} \rangle &= P^*(A=0)\Big[(\widetilde{R})_0 - (\widetilde{R}_\mathcal{F})_0\Big] + P^*(A=1)\Big[(\widetilde{R})_1 - (\widetilde{R}_\mathcal{F})_1\Big] \\ &= (1 - P^*(A=1))\Big[(\widetilde{R})_0 - (\widetilde{R}_\mathcal{F})_0\Big] + P^*(A=1)\Big[(\widetilde{R})_1 - (\widetilde{R}_\mathcal{F})_1\Big] \\ &= (\widetilde{R})_0 - (\widetilde{R}_\mathcal{F})_0 - P^*(A=1)\Big[(\widetilde{R})_0 - (\widetilde{R}_\mathcal{F})_0 - (\widetilde{R})_1 + (\widetilde{R}_\mathcal{F})_1\Big] \\ &= (\widetilde{R})_0 - (\widetilde{R}_\mathcal{F})_0 - P^*(A=1)\Big[(\widetilde{R})_0 - (\widetilde{R})_1\Big], \quad \text{since } (\widetilde{R}_\mathcal{F})_0 = (\widetilde{R}_\mathcal{F})_1. \end{aligned}$$

Finally, we conclude

$$\langle P_A^*, \widetilde{R} - \widetilde{R}_\mathcal{F} \rangle \leq 0 \quad \text{if and only if} \quad P^*(A=1) \geq \frac{(\widetilde{R})_0 - (\widetilde{R}_\mathcal{F})_0}{(\widetilde{R})_0 - (\widetilde{R})_1}.$$

$\square$

---

\*Equal Contribution.

35th Conference on Neural Information Processing Systems (NeurIPS 2021).

Next we provide a proof of Theorem 4.3 under the additional assumption that $\mathcal{A}$ and $\mathcal{V}$ are finite sets. Although less general, we feel that this proof is more instructive because it suggests the origin of (4.5).

**Theorem 4.3.** *Under the assumptions*

1. *The risk set $\mathcal{R}$ is convex.*
2. *The risk profiles with respect to $\widetilde{P}$ and $P^*$ are identical.*
3. *The unconstrained risk minimizer on unbiased data is algorithmically fair; i.e.* $\arg\min_{R \in \mathcal{R}} \langle P^*, R \rangle \subseteq \mathcal{F}_{\text{CRP}}$.

*the fair risk minimization (4.3) obtains $h \in \mathcal{H}$ such that $R(h) = R^*$ if and only if*

$$\Pi_{\mathcal{F}}(P^*_{A,V} - \widetilde{P}_{A,V}) - P^*_{A,V} \in \mathcal{N}_{\mathcal{R}}(R^*) + \mathcal{F}^{\perp}_{\text{CRP}}. \tag{1.1}$$

*where $P^*_{A,V}$ (resp. $\widetilde{P}_{A,V}$) is the marginal of $P^*$ (resp. $\widetilde{P}$) with respect to $(A, V)$, $R^*$ is the optimal risk profile with respect to $P^*$, $\mathcal{N}_{\mathcal{R}}(R^*)$ is the normal cone of $\mathcal{R}$ at $R^*$ and $\Pi_{\mathcal{F}}$ is the projection on the fair hyperplane.*

*Proof of Theorem 4.3.* **"if" direction:** Let $\widetilde{Z} = -\widetilde{P}_{A,V}$. If (4.5), then it is not hard to check that $(R^*, \widetilde{Z})$ satisfies the optimality conditions of (4.3):

$$\begin{aligned}
0 &= \widetilde{P}_{A,V} + \widetilde{Z}, \quad \text{(stationarity)} \\
R^* &\in \mathcal{F}, \quad \text{(primal feasibility)} \\
\widetilde{Z} &\in \mathcal{N}_{\mathcal{R}}(R^*) + \mathcal{F}^{\perp} \quad \text{(dual feasibility)}.
\end{aligned} \tag{1.2}$$

Indeed, we have stationarity by the definition of $\widetilde{Z}$. We have primal feasibility because the unconstrained risk minimizer on unbiased data is algorithmically fair: $R^* \in \mathcal{F}$. We have dual feasibility because

$$\begin{aligned}
\widetilde{Z} &= \Pi_{\mathcal{F}_{\text{CRP}}}(P^*_{A,V} - \widetilde{P}_{A,V}) - P^*_{A,V} + \Pi_{\mathcal{F}^{\perp}_{\text{CRP}}}(P^*_{A,V} - \widetilde{P}_{A,V}) \\
&\in \mathcal{N}_{\mathcal{R}}(R^*) + \mathcal{F}^{\perp}_{\text{CRP}} + \mathcal{F}^{\perp}_{\text{CRP}} \\
&= \mathcal{N}_{\mathcal{R}}(R^*) + \mathcal{F}^{\perp}_{\text{CRP}},
\end{aligned}$$

where we appealed to (4.5) in the second step and recalled $\mathcal{F}^{\perp}_{\text{CRP}}$ is a subspace in the third step. The FRM problem (4.3) is convex, so (1.2) implies $R^*$ is an optimal point of (4.3).

**"only if" direction:** Assume $R^*$ solves (4.3). This implies there is $\widetilde{Z} \in \mathcal{N}_{\mathcal{R}}(R^*) + \mathcal{F}^{\perp}_{\text{CRP}}$ such that $(R^*, \widetilde{Z})$ satisfies (1.2). By the stationary and dual feasibility conditions,

$$\widetilde{Z} = -\widetilde{P}_{A,V} \in \mathcal{N}_{\mathcal{R}}(R^*) + \mathcal{F}^{\perp}_{\text{CRP}}.$$

We write $\widetilde{P}_{A,V}$ as $\Pi_{\mathcal{F}_{\text{CRP}}}(P^*_{A,V} - \widetilde{P}_{A,V}) - P^*_{A,V} + \Pi_{\mathcal{F}^{\perp}_{\text{CRP}}}(P^*_{A,V} - \widetilde{P}_{A,V})$ and rearrange to obtain

$$\begin{aligned}
\Pi_{\mathcal{F}_{\text{CRP}}}(P^*_{A,V} - \widetilde{P}_{A,V}) - P^*_{A,V} &\in \Pi_{\mathcal{F}^{\perp}_{\text{CRP}}}(P^*_{A,V} - \widetilde{P}_{A,V}) + \mathcal{N}_{\mathcal{R}}(R^*) + \mathcal{F}^{\perp}_{\text{CRP}} \\
&= \mathcal{N}_{\mathcal{R}}(R^*) + \mathcal{F}^{\perp}_{\text{CRP}},
\end{aligned}$$

where we recalled $\mathcal{F}_{\text{CRP}}$ is a subspace in the second step. $\square$

**Corollary 4.4.** *A sufficient condition for (4.5) is $\widetilde{P}_{A,V} - P^*_{A,V} \in \mathcal{F}^{\perp}_{\text{CRP}}$.*

*Proof of Corollary 4.4.* If $\widetilde{P}_{A,V} - P^*_{A,V} \in \mathcal{F}^{\perp}_{\text{CRP}}$, then $\Pi_{\mathcal{F}_{\text{CRP}}}(P^*_{A,V} - \widetilde{P}_{A,V}) = 0$, so we need to check that $-P^*_{A,V} \in \mathcal{N}_{\mathcal{R}}(R^*) + \mathcal{F}^{\perp}_{\text{CRP}}$. For any $R \in \mathcal{R}$,

$$\langle -P^*_{A,V}, R - R^* \rangle = \langle P^*_{A,V}, R^* - R \rangle \leq 0$$

as $R^*$ is the minimizer of $\langle P^*_{A,V}, R \rangle$ over $\mathcal{R}$. This shows $-P^*_{A,V} \in \mathcal{N}_{\mathcal{R}}(R^*)$ as desired. $\square$

## 2 Continuous discriminative attributes

In this section, we state and prove a more general verion of Theorem 4.3 that permits continuous discriminative attributes. In this more general setting, risk profiles are (integrable) functions on $\mathcal{Z} \triangleq \mathcal{A} \times \mathcal{V}$, so the fair risk minimization problem (1.2) and its unconstrained counterpart are infinite dimensional optimization problems. We start by setting up the problem and reviewing relevant results from optimization theory.

Let $(\mathcal{Z}, \Sigma)$ be a measurable space and $\mathcal{S}$ be the set of bounded measurable functions on $(\mathcal{Z}, \Sigma)$. We equip $\mathcal{S}$ with the sup norm. The risk set $\mathcal{R}$ and the fair constraint set $\mathcal{F}$ are generally subsets of $\mathcal{S}$. The (topological) dual of $\mathcal{S}$ (denoted by $\mathcal{S}'$) is the set of finitely additive measures on equipped with the total variation norm [3]. This result allows us to represent continuous linear functionals on such spaces with (finitely additive) measures, so it is a generalization of the more familiar Riesz–Markov–Kakutani representation theorem to spaces of (possibly discontinuous) measurable functions. We observe that the more familiar set of countably additive measures is a closed subset of $\mathcal{Z}'$.

**Definition 2.1** (Complemented subspace). *Let $\mathbb{B}$ be a Banach space and $A \subset \mathbb{B}$ be a subspace. We say $A$ is complemented subspace of $\mathbb{B}$, if there exists another subspace $A_C \subset \mathbb{B}$ such that $\mathbb{B} = A \oplus A_C$.*

Henceforth, for if $A$ is a complemented subset of a Banach space $\mathbb{B}$, (*i.e.*, $A \oplus A_c = \mathbb{B}$) then we define $\Pi_{A,A_C}(x)$ (resp. $\Pi_{A_C,A}(x)$) is the component of $x$ in $A$ (resp. $A_C$), i.e. $\Pi_{A,A_C}(x) = x_1$ (resp. $\Pi_{A_C,A}(x) = x_2$) where $x = x_1 + x_2$ with $x_1 \in A, x_2 \in A_C$. Recall that, we define $\mathcal{F}$ as the fair hyperplane. Previously it was a subspace of the risk set, now it becomes a subspace of $\mathcal{S}$. We have the following assumption on the fair hyperplane:

**Definition 2.2** (Annihilator). *For any $A \subset \mathbb{B}$ we define its annihilator $A^\perp \subset \mathbb{B}'$ as the set of bounded linear functions $f : \mathbb{B} \to \mathbf{R}$ such $f(x) = 0$ for all $x \in A$.*

**Lemma 2.3.** *Let $A$ be a complemented subspace in $\mathbb{B}$. Then $A^\perp$ is complemented in $\mathbb{B}'$.*

*Proof.* Since, $A$ is complemented in $\mathbb{B}$, there exists a subspace $G \subset \mathbb{B}$ such that $A \oplus G = \mathbb{B}$. This implies, each $x \in \mathbb{B}$ has the unique decomposition $x = x_1 + x_2$, where $x_1 \in A$ and $x_2 \in G$. We consider the projection ma p $\Pi_{A,G} : \mathbb{B} \to \mathbb{B}$ such that $\Pi_{A,G}(x) = x_1$. Let us define two following subspaces in $\mathbb{B}'$ :

$$\mathcal{H}_{A,G} = \{f \circ \Pi_{A,G} \mid f \in \mathbb{B}'\}$$
$$\bar{\mathcal{H}}_{A,G} = \{f - f \circ \Pi_{A,G} \mid f \in \mathbb{B}'\}.$$

Note that, $\bar{\mathcal{H}}_{A,G} \subset A^\perp$. Also, for any $f \in A^\perp$ we have $f \circ \Pi_{A,G} = 0_{\mathbb{B}'} \implies f = f - f \circ \Pi_{A,G} \in \bar{\mathcal{H}}_{A,G}$. This implies $\bar{\mathcal{H}}_{A,G} = A^\perp$. Furthermore, $\mathbb{B}' = \mathcal{H}_{A,G} + \bar{\mathcal{H}}_{A,G}$ and for any $f \in \mathcal{H}_{A,G} \cap \bar{\mathcal{H}}_{A,G}$ we have $f(A) = f(G) = \{0\}$. Hence, $f = 0_{\mathbb{B}'}$. This implies $\mathbb{B}' = \mathcal{H}_{A,G} \oplus \bar{\mathcal{H}}_{A,G} = \mathcal{H}_{A,G} \oplus A^\perp$. $\square$

Finally, we review some relevant background on infinite dimensional optimization. Since we are mostly concerned with convex optimization problems with linear cost functions, the theory simplifies considerably.

**Definition 2.4** (tangent cone). *The tangent cone of a closed convex set $\mathcal{C} \subset \mathbb{B}$ at a point $x \in \mathcal{C}$ is the closure of the cone of feasible directions at $x$:*

$$T_{\mathcal{C}}(x) \triangleq \mathsf{cl}\{d \in \mathbb{B} \mid \text{there is } \bar{t} > 0 \text{ such that } x + td \in \mathcal{C} \text{ for all } t \in [0, \bar{t}]\}.$$

There are many notions of tangent cone in variational analysis (*e.g.* Clarke tangent cone, contingent cone, inner tangent cone *etc.*), but they all coincide for closed convex sets [2]. Notably, this definition is identical to the definition (for convex sets) in finite dimensions.

**Definition 2.5** (normal cone). *The normal cone of a closed convex set $\mathcal{C} \subset \mathbb{B}$ at a point $x \in \mathcal{C}$ is the polar cone of the tangent cone of $\mathcal{C}$ at $x$:*

$$N_{\mathcal{C}}(x) \triangleq \{d' \in \mathbb{B}' \mid \langle d', d \rangle \leq 0 \text{ for all } d \in T_{\mathcal{C}}(x)\}.$$

**Proposition 2.6.** *Let $\mathcal{C}$ be a closed convex subset of a Banach space $\mathbb{B}$. Consider the convex optimization problem*

$$\min_{x \in \mathcal{C}} \langle c, x \rangle.$$

*A point $x^* \in \mathcal{C}$ is an optimal point iff*

$$\langle c, d \rangle \geq 0 \text{ for any } d \in T_{\mathcal{C}}(x^*),$$

*where $\langle c, \cdot \rangle$ is the linear cost function and $T_{\mathcal{C}}(x^*)$ is the tangent cone of $\mathcal{C}$ at $x^*$. Equivalently, $x^*$ is optimal if and only if $c \in N_{\mathcal{C}}(x^*)$.*

Recall that in a normed vector space $\langle f, x \rangle$ means the value of the linear functional $f$ at $x$. In our problem setting, points in the normed space $\mathcal{S}$ are integrable functions/random variables and linear functionals on $\mathcal{S}$ are (finitely additive) measures, so $\langle f, x \rangle$ means expectation of the random variable x with respect to probability measure $f$.)

We are ready to state the extension of our main result to continuous discriminative attributes. Assumptions in Theorem 4.3 from the main paper remain in effect. For continuous discriminative attributes, we impose an additional assumption.

**Assumption 2.7.** *The fair subspace $\mathcal{F}$ is complemented in $\mathcal{S}$.*

This assumption is usually satisfied by common algorithmic fairness constraints: when RP is considered, $\mathcal{F}$ is the set of all constant functions from $\mathcal{A}$ to $\mathbf{R}$. For CPR, $\mathcal{F} \subseteq \mathcal{S}$ is the set of all functions $f : \mathcal{A} \times \mathcal{Y} \to \mathbf{R}$ such that $f$ is constant on the first co-ordinate, i.e. $f(x_1, y) = f(x_2, y)$ for all $x_1 \neq x_2 \in \mathcal{A}$ and $y \in \mathcal{Y}$. We now argue that, in both the cases $\mathcal{F}$ is a complemented subset of $\mathcal{S}$ under mild assumptions. For RP, we use the fact that any subspace $A \subseteq \mathcal{S}$ with $\dim(A) < \infty$ or $\mathrm{codim}(A) < \infty$ is complemented. As $\mathcal{F}^{RP}$ is the set of all constant functions, it has dimension 1 and hence complemented. For CRP, assume that there exists some base measure $\mu$ such that $f \in \mathcal{S}$ is integrable with respect to $\mu$. Then one can write: $f = f_1 + f_2$ where $f_1 \in \mathcal{F}$ which is defined as: $f_1(a, v) = g(v)$ where $g(v)$ is the marginal of $f(\cdot, v)$ with respect to the base measure $\mu$. The function $f_2$ is analogously defined as $f - f_1 \equiv f(a, v) - g(v)$.

**Theorem 2.8.** *If the unconstrained risk minimizer on unbiased data is algorithmically fair (i.e. its risk profile $R^*$ satisfy the fairness constraints), then fair risk minimization (4.3) learns $h \in \mathcal{H}$ such that $R(h) = R^*$ under 2.7 and convexity of $\mathcal{R}$ if and only if*

$$\Pi_{\mathcal{F}_C^\perp, \mathcal{F}^\perp}(P_{A,V}^* - \widetilde{P}_{A,V}) - P_{A,V}^* \in \mathcal{N}_{\mathcal{R}}(R^*) + \mathcal{F}^\perp. \tag{2.1}$$

*where $P_{A,V}^*$ (resp.$\widetilde{P}_{A,V}$) is the marginal of $P^*$ (resp. $\widetilde{P}$) with respect to $(A, V)$, $\mathcal{N}_{\mathcal{R}}(R^*)$ is the normal cone of $\mathcal{R}$ at $R^*$ and $\Pi_{\mathcal{F}_C^\perp, \mathcal{F}^\perp}(\cdot)$ is the projection as defined previously.*

*Proof.* For notation simplicity define $X \triangleq \Pi_{\mathcal{F}_C^\perp, \mathcal{F}^\perp}(P_{A,V}^* - \widetilde{P}_{A,V}) - P_{A,V}^*$. We show that $\min_{R \in \mathcal{F}} \langle \widetilde{P}, R \rangle = \langle \tilde{P}, R^* \rangle$ holds if and only if $X \in \mathcal{N}_{\mathcal{R}}(R^*)$. Towards that end, fix $R \in \mathcal{F}$:

$$
\begin{aligned}
\langle \widetilde{P}, R \rangle &= \langle \widetilde{P} - P^*, R \rangle + \langle P^*, R \rangle \\
&= \langle \Pi_{\mathcal{F}_C^\perp, \mathcal{F}^\perp}(\widetilde{P} - P^*), R \rangle + \langle P^*, R \rangle \\
&= \langle -P^* - X, R \rangle + \langle P^*, R \rangle \\
&= \langle -X, R \rangle \\
&= \langle -X, R^* \rangle + \langle -X, R - R^* \rangle \\
&= \langle \tilde{P}, R^* \rangle + \langle -X, R - R^* \rangle \quad \text{[From equation (2.2)]}
\end{aligned} \tag{2.2}
$$

Hence we have: $\min_{R \in \mathcal{F}} \langle \widetilde{P}, R \rangle = \langle \tilde{P}, R^* \rangle$ if and only if $\langle -X, R - R^* \rangle \geq 0$ for all $R \in \mathcal{F}$ which holds if and only if $X \in \mathcal{N}_{\mathcal{R} \cap \mathcal{F}}(R^*) = \mathcal{N}_{\mathcal{R}}(R^*) + \mathcal{F}^\perp$. This completes the proof. $\square$

# 3 Experimental details

We provide additional details to help reproduce our results. Please also see the code provided with the submission. Code for the Reductions classifier [1] is available here: `https://github.`

`com/fairlearn/fairlearn`. We modified the source code to prevent it from early stopping, so the baseline classifier runs for same number of iterations as the fair classifier. The idea behind the Reductions approach is to translate the problem of learning a fair classifier into a constraint optimization problem, where constraints depend on the fairness definition of choice. Reductions method requires a base classifier: it learns an ensemble of the base classifiers to optimize performance subject to the fairness constraints. We used logistic regression as the base classifier in all experiments. The other important parameter is the tolerance $\epsilon$ that controls the amount of permissible constraint violation. Smaller tolerance implies tighter fairness constraints. In the Figure 1 experiment we used Demographic Parity fairness constraint, and in all other experiments we used Equalized Odds fairness constraint [4] with $\epsilon = 10$ for the baseline classifier (i.e. fairness can be arbitrarily violated) and $\epsilon = 0.1$ (for the Adult experiment $\epsilon = 0.02$) for the fair classifier.[1]

**Simulations** Simulated data is generated from $X|A = a, Y = y \sim \mathcal{N}(\mu_{ay}, \Sigma_{ay})$ in 2-dimensions with prescribed $A, Y$ joint distribution. We fixed $Y$ marginals $p_{\cdot 0} = p_{\cdot 1} = 0.5$ and varied joint $P_{A,Y}$ (in Figure 1 for the test data, setting train data values $p_0 = 0.4$ and $p_1 = 0.1$; and in Figures 4 and 5 for the train data, setting test data values $p_{\cdot\cdot} = 0.25$) to study different degrees of label bias. Reductions was trained for 25 iterations for both baseline and fair classifiers. We provide code reproducing Figure 5 of the main text in `simulations.py`. Please also refer to the code for concrete values of $\{\mu_{ay}, \Sigma_{ay}\}$ and other minor details.

**COMPAS experiment** Reductions was trained for 50 iterations for both baseline and fair classifiers. We provide code reproducing one run of the experiment for Table 1 of the main text (results in the table summarize 100 runs) in `compas.py`. Please also refer to the code for data pre-processing and other minor details.

**Adult experiment** We ran experiment on the Adult dataset with the same setup as in the COMPAS experiment. We summarize results over 100 runs in Table 1.

Table 1: Accuracy on Adult data

|      | $P^*$ | $\widetilde{P}$ |
|------|-------|-------|
| Fair | **0.852**$\pm$0.004 | 0.843$\pm$0.003 |
| Base | 0.848$\pm$0.005 | **0.847**$\pm$0.003 |