# OpenReview forum: "Does enforcing fairness mitigate biases caused by subpopulation shift?"
_NeurIPS.cc/2021/Conference — NeurIPS 2021 Poster_

### Official Review · Reviewer_VZHD · 2021-07-08

**Rating:** 5
**Confidence:** 3

**Summary:**

The paper studies the impact of constrained fairness risk minimization on model performance when data are affected by subpopulation shifts. The authors focus on two notions of fairness that are defined with respect to the loss of the model, and that they call "risk parity" and "conditional risk parity" respectively.
Under a series of assumptions, the authors derive the conditions under which a model model trained via constrained risk minimization has overall higher/lower performance on the target data (vs. train data). They then examine the resulting bounds via simulations and experiments.

**Limitations And Societal Impact:**

The authors should clearly motivate how their results advance and could contribute to our understanding of risk minimization with fairness constraints, e.g., perhaps by referring to the their case study of the COMPAS data.

**Main Review:**

My comprehension of the results was very much hindered by the lack of clearly defined notation (see suggestions below). I also had a hard time understanding how these results differ from prior work and the meaning of the bounds in the results. Two important points:
* It seems that assumption (3.3) is fairly limiting. For example, this assumption rules out the setting of label-bias as defined by [1] and thus . In addition, I looked at the paper in reference [16] (line 92), and I have the impression that (3.3) is an additional assumption imposed on the subpopulation shifts setting, rather than being “(slightly) less restrictive”.
* That said, it is unclear to me why the bounds per se, in their current form, should be of interest, or at least I do not think that the result in theorem 3.2 was properly discussed. We should probably interested in understanding by how much performance will change, although I am not sure this is analytically possible.

Some (hopefully) constructive suggestions are: (i) move related work up and clearly explain how this work fits into the existing literature, (ii) clearly define all notation, (iii) improve the explanations of the results, (iv) add an organization paragraph in the introduction.

Some of the typos and other minor issues:
* definition 3.1: is this definition based on prior work?
* line 36 “during improves”
* line 63: needs citation and \mathcal{l} has never been introduced
* line 65: the original study should be cited
* ERM has never been defined
* (3.2) I am not sure that the inner product on the RHS makes sense
* line 77: erroneous reference (4.3)
* it’s unclear what the pedix in the expectation means (compare pedix in (3.1) vs. Line 76)
* (3.2) \mathcal{F} has never been defined
* line 75: the bound between 0 and 1 for \tilde{P}_A makes sense only in case of discrete distributions
* (3.3) the expectations E* and \tilde{E} have not been defined
* (4.4) V has never been properly defined
* example 4.2: I believe that the definition of label bias adopted by the authors is one of many, so they may want to cite a paper the authors took this definition from. For example, [1] calls (4.2) “under-representation and labeling bias” (I see that this paper is actually cited in the related work).
* line 185: “even if the Bayes decision rule in the target domain is algorithmically fair” is unclear
* line 147: “.” Should be “,”

[1] Blum, Avrim, and Kevin Stangl. "Recovering from biased data: Can fairness constraints improve accuracy?." arXiv preprint arXiv:1912.01094 (2019).

**Time Spent Reviewing:**

7

---

> ### Author Response · Authors · 2021-08-10
> **Response**
>
> We thank you for the review. Please see our responses below.
>
> **It seems that assumption (3.3) is fairly limiting. For example, this assumption rules out the setting of label-bias as defined by [1] and thus. In addition, I looked at the paper in reference [16] (line 92), and I have the impression that (3.3) is an additional assumption imposed on the subpopulation shifts setting, rather than being “(slightly) less restrictive”.**
>
> Assumption (3.3) is a particular instance of subpopulation shift, in which the subpopulations are defined as all individuals sharing a particular value of the sensitive attribute. In section 4, we consider more general instances of subpopulation shift. The high level idea of subpopulation shift is that the train and test distribution differ in "how well-represented each subpopulation is" (e.g. [2], Section 2). This means the each subpopulation stays same across train and test data, but the proportions of subpopulations that make up the train and test data differ. In other words, the train and test data and mixtures of the same subpopulations; only the mixture weights are different. Mathematically speaking, for RP subpopulations are created by conditioning on different values of $A$ and for CRP by conditioning on different values of ($A$, $V$). This means the conditional distributions with respect to a particular value of $A$ (for RP) or ($A, V$) (for CRP) are same for both training and test data distribution, whereas the marginals (w.r.t $A$ for RP and w.r.t $(A, V)$ for CRP) may vary.
>
> In equation (3.3) or (4.4), we do not require the conditional distribution to be same, only the conditional expectations of the loss to be the same. This is why we say equation (3.3) or (4.4) is (slightly) less restrictive than the usual subpopulation shift assumption.
>
> Although label-bias defined in Blum and Stangl is a specific instance of distributional shift that is not covered by subpopulation shift, subpopulation shift is a broadly applicable assumption in domain adaptation. There are benchmark datasets dedicated to this setting [1, 2]. In addition, subpopulation shift has been identified as a source of many instances of algorithmic bias [3, 4, 5].
>
> **That said, it is unclear to me why the bounds per se, in their current form, should be of interest, or at least I do not think that the result in theorem 3.2 was properly discussed. We should probably interested in understanding by how much performance will change, although I am not sure this is analytically possible.**
>
> The change in performance of the classifier upon enforcing fairness constraint depends both on "the shape of risk profile" and "how training distribution differs from the test distribution". As you have correctly mentioned, it is difficult to quantify analytically the change in performance of a classifier without further assumptions on the risk profiles or on the relation between training and test distribution. This paper takes a step towards this direction: in Theorem 3.2 and 4.3, we provide a characterization on the relation between training and test distribution (based on certain assumptions on the risk profiles) under which it is possible to improve the accuracy on the test distribution via enforcing fairness constraint during training.
>
>
> **My comprehension of the results was very much hindered by the lack of clearly defined notation.**
>
> We thank you for pointing out typos and suggesting improvements for notation, which we will incorporate in the revised version. However, we note that other reviewers found the presentation and notations sufficiently clear, e.g. "The notation is clear" (3vrA), "The paper is structured clearly and is well-written!" (3vrA), "The writing is clear." (jgBW), "This paper is well-written" (M7Vx).
>
>
> **definition 3.1: is this definition based on prior work?**
>
> The idea of RP is closely related to demographic parity where the model outcome is assumed to be independent of $A$. The risk parity assumes similar condition in terms of the loss function.
>
> **$\mathcal{l}$ has never been introduced**
>
> $\mathcal{l}$ is the loss function. We'll define it in a revision.
>
> **ERM has never been defined**
>
> Here ERM means empirical risk minimization.
>
> **I am not sure that the inner product on the RHS makes sense.**
>
> As $A$ takes value on the discrete set and $\tilde{P}_A$ is the marginal probability mass function of $A$ for the training set, it is a vector in $\mathbf{R}^{|A|}$. Similarly, $R$ is the risk profile under under $\tilde P$, i.e. the conditional expectation of $\ell(h(X), Y)$ given $A$. Therefore it is also a vector in $\mathbf{R}^{|A|}$. Hence the inner product in (3.2) is simply the $\ell_2$-inner product in $\mathbf{R}^{|A|}$.
>
> **It’s unclear what the pedix in the expectation means (compare pedix in (3.1) vs. Line 76)**
>
> We are responding assuming that "pedix" means tilde. $\tilde{E}_A$ in line 76 refers to expectation with respect to $\tilde{P}_A$, the marginal distribution of $A$ in the source/training domain. The expectation in Equation (3.1) is the conditional expectation of loss, where the condition is being done on different values of $A$.
>
> **(3.2) $\mathcal{F}$ has never been defined**
>
> $\mathcal{F}$ here is same as $\mathcal{F}_{RP}$. We will update the notation in the revised version of the paper.
>
> **Line 75: the bound between 0 and 1 for $\tilde{P}_A$ makes sense only in case of discrete distributions.**
>
> As $A$ is a sensitive attribute, it is typically assumed to be discrete.
>
> **(3.3) the expectations $E^\*$ and $\tilde{E}$ have not been defined.**
>
> Here $\tilde E$ (resp. $E^*$) denotes the expectation with respect to $\tilde P$ (resp. $P^*$).
>
> **(4.4) V has never been properly defined**
>
> It is defined in line 129.
>
> **Example 4.2: I believe that the definition of label bias adopted by the authors is one of many, so they may want to cite a paper the authors took this definition from. For example, [1] calls (4.2) “under-representation and labeling bias” (I see that this paper is actually cited in the related work)**
>
> We agree that the definition of label bias in Example 4.2 differs from that in Blum and Stangl. To avoid confusion, we will not call Example 4.2 label bias in a revision.
>
> **Line 185: “even if the Bayes decision rule in the target domain is algorithmically fair” is unclear**
>
> The precise meaning of this statement is defined in the equation following immediately after this statement.
>
> ---
> References
>
> [1] Koh, Pang Wei, et al. Wilds: A benchmark of in-the-wild distribution shifts, 2021.
>
> [2] Santurkar, Shibani, Dimitris Tsipras, and Aleksander Madry. Breeds: Benchmarks for subpopulation shift, 2020.
>
> [3] Buolamwini, Joy, and Timnit Gebru. Gender shades: Intersectional accuracy disparities in commercial gender classification, 2018.
>
> [4] Francisco, M., and Carlos D. Bustamante. Polygenic risk scores: a biased prediction? 2018.
>
> [5] Sjoding, Michael W., et al. Racial bias in pulse oximetry measurement, 2020.

---

> > ### Comment · Reviewer_VZHD · 2021-08-22
> > **Discussion**
> >
> > I want to thank the authors for their thoughtful response. I have carefully
> > gone through that, as well as through the other reviewers' comments and the
> > paper again. I have substantially increased my score to reflect my improved
> > understanding of the paper in light of the authors' responses. However, I still
> > have concerns around its readability and the communication of the key message in
> > the paper.
> >
> > I am glad to see that other reviewers found the paper to be well written. I
> > think that the typos, some of which I pointed out in my review, could be easily
> > fixed in the final version of the paper. However,
> > *   I think
> > that the term "subpopulation shift" should be defined mathematically, as the
> > authors did in their response.
> > *  notation is sometimes overloaded. Both in
> > their response and in the paper, the authors use $ \tilde P_{A} $ to indicate the
> > vector containing the different values of the marginal probability mass function
> > of $A$. Obviously, if $\tilde P_{A}$ is indeed a
> > vector, then the inner product makes sense. However,
> > I find this to be very confusing because in lines 50-51 the notation
> > $P^*$ and $\tilde P$ is used to indicate densities (this was unclear as well
> > given that they could have well been c.d.f.'s) which are clearly not vectors. Similarly, I was confused by the
> > notation used for the expectation. Apologies about the use of the term "pedix"
> > which does not really appear in any dictionary! I used the term to refer to the
> > subscript. Please compare equation (3.1) and line 76. In equation (3.1), the
> > authors have used the notation $E_{\tilde P}$ to indicate the expectation taken
> > over $\tilde P$. Why not use $E_{\tilde P_A}$ in line 75 then?
> > * it should
> > be made clear that $\mathcal{A}$, as presented in the main body of the paper, is
> > a discrete set. I understand that results have been generalized to the
> > continuous case in the supplementary material.
> > * despite most people know
> > that the acronym "ERM" stands for "empirical risk minimization", the acronym
> > should still be defined, I believe. That was the point of my comment in the review, I was not asking for its definition.
> > *  Additional
> > typos that I found while rereading the paper: (i) in line 109, "DP" should
> > probably be "RP"; (ii) lines 212-213 contain two typos; (iii) another typo is present in
> > line 245; (iv) in line 330, "column" may not be the right term; (v) in line
> > 345, COMPAS data contain information on whether the defendants were rearrested,
> > not about whether they reoffended or not. Offenses may not result in arrests. This distinction is important.
> >
> >
> > Regarding the content of the paper, the authors' answer to one of the questions
> > asked by reviewer jgBW helped me get more clarity on the utility of the bounds.
> > Now I start to see how the result in theorem 3.2 could be used in practice. In
> > the current version of the paper, however, this crucial discussion is missing
> > and it should definitely be added. It should also be supported by experimental
> > results, which are currently missing. Theorem 4.3 seems to have limited
> > applicability given that assumption 3 cannot be verified and seems rather
> > unrealistic. Yet it still is an interesting result. Additional comments:
> > * Since the definition of subpopulation shift used in this
> > paper was not adopted from prior work, it should probably be motivated.
> > * I
> > read example 4.2 again and the setting described is actually referred to
> > as "under-representation bias" by Blum and Stangl. That said, I still have a
> > hard time understanding how the results in this paper "complement" (line 282)
> > those in Blum and Stangl's paper for label bias. If they do, then this should be
> > properly discussed. If they do not, then the related work section should be changed
> > accordingly, e.g., by dropping the reference to this type of bias.
> > * Since
> > assumption 3 in theorem 4.3 cannot be checked, I am not sure about why the
> > results observed on COMPAS should be expected. As I mentioned above, perhaps
> > it'd be more helpful to show an example where the result in theorem 3.2 is used
> > in practice.

---

> > > ### Author Response · Authors · 2021-08-23
> > > **Response to Reviewer VZHD comments**
> > >
> > > We thank the reviewer for their time and effort, and for additional comments. We will incorporate the suggested changes concerning notations, typos, and abbreviations in the final version of the paper.
> > >
> > > **Now I start to see how the result in theorem 3.2 could be used in practice. In the current version of the paper, however, this crucial discussion is missing and it should definitely be added. It should also be supported by experimental results, which are currently missing.**
> > >
> > > We are glad that our discussion helped to clarify the practical use of our results. We will add this discussion into the final version of the paper. Regarding the experiments, our COMPAS and Adult experiments are instances of application of theorem 3.2 in practice (see our response to the last bullet point regarding the COMPAS experiment). We will extend the discussion in the empirical section to make the connection clearer.
> > >
> > > **Since the definition of subpopulation shift used in this paper was not adopted from prior work, it should probably be motivated.**
> > >
> > > We will describe distribution definition of subpopulation-shift from [2] and use it to motivate our risk-based definition of subpopulation shift.
> > >
> > > **I read example 4.2 again and the setting described is actually referred to as "under-representation bias" by Blum and Stangl. That said, I still have a hard time understanding how the results in this paper "complement" (line 282) those in Blum and Stangl's paper for label bias. If they do, then this should be properly discussed. If they do not, then the related work section should be changed accordingly, e.g., by dropping the reference to this type of bias.**
> > >
> > > We thank the reviewer for taking additional time to look at both papers. The reviewer is correct that Example 4.2 is "under-representation bias" in [1]. In this paper, we focus on subpopulation shift. Under-representation bias by [1] is an instance of subpopulation shift, but label bias by the same authors is not an instance of subpopulation shift. Thus our results complement the results in [1] for under-representation bias (as they provide sufficient conditions whereas we provide necessary and sufficient conditions), but our results do not apply to the label bias setting in Blum and Stangl. We will clarify this in the the related work section.
> > >
> > > **Since assumption 3 in theorem 4.3 cannot be checked, I am not sure about why the results observed on COMPAS should be expected. As I mentioned above, perhaps it'd be more helpful to show an example where the result in theorem 3.2 is used in practice.**
> > >
> > > The test data in COMPAS experiment is subsampled to satisfy the sufficient condition in line 259-260 under conditional risk parity. Note that, in line 347-349, $P^*$ has same marginal of $Y$ as in $\tilde P$. Furthermore, $P^*$ has equal representation for each of the 4 values of the protected attribute within each class ($Y = 0$ or 1). This implies the column sums of $P^*_{A,Y}-\tilde P_{A,Y}$ are 0, as required by the sufficient condition in line 260.
> > >
> > > Regarding assumption 3, equal representation of the protected groups in $P^*$ is sufficient for satisfying the assumption under an additional condition that noise levels are similar across protected groups (i.e. classifiers fit for each protected group independently have similar accuracy on their respective protected groups - this is easy to check on the train data). We will clarify this in the revision. This does serve as an example of theorem 3.2 used in practice: if we expect equal representation of the protected groups at test time (and check for the similarity of noise levels on the train data), we should expect that enforcing fairness will improve accuracy.
> > >
> > > ---
> > > References
> > >
> > > [1] Blum, Avrim, and Kevin Stangl. Recovering from biased data: Can fairness constraints improve accuracy?.
> > >
> > > [2] Santurkar, Shibani, Dimitris Tsipras, and Aleksander Madry. Breeds: Benchmarks for subpopulation shift, 2020.

---

### Official Review · Reviewer_M7Vx · 2021-07-15

**Rating:** 7
**Confidence:** 4

**Summary:**

This paper addresses the following problem: What will the performance of a ML model be on the target domain if one enforces fairness in the training space, while considering the fact that some groups may be underrepresented in the training domain (and hence, biased) and considering subpopulation shift? The paper brings theoretical results as well as experiments to provide insights into this phenomenon.

**Limitations And Societal Impact:**

See my main review.

**Main Review:**

This paper is well-written and well-motivated. The problem statement and the contributions of the paper are clear. Specially, I appreciate the {\em insight} that this paper can bring to the community. One of the main points of the paper is characterizing and analyzing algorithmic fairness under subpopulation shifts, which is defined in the paper.

The theoretical results of the paper are backed by the experiments on the simulated data, as well as COMPAS and Adult data.

Note: One should be careful with interpreting results of this paper as the paper assumes the assumption in equation (3.3). This is assumption 2 under Theorem 3.2 and 4.3. It means that this paper does not consider an entire distribution shift from the train domain to the target domain and restricts its attention to subpopulation shift. Hence, one should be cautious about making general statements based on this paper. Perhaps, a discussion about this can help the general reader more.

Minor: Contribution number 3 in Introduction: Line 36, please complete the sentence.


**Time Spent Reviewing:**

48

---

> ### Author Response · Authors · 2021-08-10
> **Response**
>
> We thank the reviewer for the feedback. We have incorporated the minor comment in the revised version of the paper. Please see our response to the other comment below.
>
>
> **One should be careful with interpreting results of this paper as the paper assumes the assumption in equation (3.3). This is assumption 2 under Theorem 3.2 and 4.3. It means that this paper does not consider an entire distribution shift from the train domain to the target domain and restricts its attention to subpopulation shift. Hence, one should be cautious about making general statements based on this paper. Perhaps, a discussion about this can help the general reader more.**
>
> Thank you for your comment. Yes, we are only operating under subpopulation shift, and other forms of general distribution shift are not considered in this paper. However, this is a fruitful area for future work, which we will mention in the summary and discussion section of the revised version of this paper.

---

### Official Review · Reviewer_3vrA · 2021-07-15

**Rating:** 7
**Confidence:** 2

**Summary:**

The authors analyze the effect of enforcing algorithmic fairness (performance parity) in the presence of subpopulation shift. They argue that model performance can be summarized in terms of risk profiles, as risk minimization problems are equivalent to linear optimization wrt risk profiles. Basically, the authors consider ML problems that can be mapped to risk minimization. The paper also provides a way to analyze subpopulation shift by decomposing the training bias into (1) a portion which is orthogonal to the fair constraint, and (2) a robustness of $R^\star$ to changes in the distribution of the data in the hypothetical target domain ($P^\star$). Lastly, the authors show that fair risk minimization while training does not necessarily improve performance---a theoretical analysis is provided that shows when improvement and harm is done to model performance. Computational results are shown over synthetic data and recidivism data (COMPAS) to support results.

**Limitations And Societal Impact:**

Yes.

**Main Review:**

ORIGINALITY: The main contribution of this paper is to offer insight and an interesting perspective into bias mitigation caused by population shift. The problem is well-motivated; this is not an uncommon phenomenon in practice. The work seems sufficiently original, although I am not intimate with the frontier of knowledge in this area.  There are already works that evaluate whether FRM reduces/impacts algorithmic bias. One key difference is that the authors are looking to evaluate these criteria in the target domain. The related work seems thorough in differentiating between the contributions of this work and similar works.
QUALITY: The technical content of the main paper appears correct---I thank the authors for the experimental details in the main text. The optimization review in the Appendix B adds to the general readability/understanding of the techniques used in the proof. I did not look over the supplementary proofs in A but focused on B.8, the permissibility of continuous-valued attributes.
CLARITY: The paper is structured clearly and is well-written! The notation is clear and intuition is provided for different concepts, e.g., enforcing RP by solving the constrained risk minimization problem using two groups...visual explanation is provided and clearly explained. The goal of the computational results is explained clearly, and adequate experimental details are provided in the main text.
SIGNIFICANCE: I enjoyed this paper. It sheds light on a well-known phenomenon seen in practice---distribution (subpopulation) shift, and could be useful as a tool for analyzing/creating fairness aware algorithms. I believe this work adequately expands on previous work, e.g., ML problems applicable to risk minimization and considering posterior drift, while focusing on the effect on the target domain.

**Time Spent Reviewing:**

7

---

> ### Author Response · Authors · 2021-08-10
> **Response**
>
> We thank you for the positive and encouraging feedback! Kindly let us know if there is anything you would like us to discuss.

---

### Official Review · Reviewer_jgBW · 2021-07-17

**Rating:** 6
**Confidence:** 4

**Summary:**

The work examines when does a fairness constrained model improve in overall accuracy on a target dataset which has a different distribution (of loss conditioned on some covariates) from the train dataset. It provides necessary and sufficient conditions under which this happens. Experiments on synthetic data validate the theoretical results. Real world experiments on COMPASS and Adult income datasets are promising.

**Ethical Concerns:**

The authors have addressed the ethical concern posed by fair learning without attending to differences between training and deployment settings.

**Limitations And Societal Impact:**

Please discuss limitations of the work e.g. limited to two fairness metrics (risk parity and conditional risk parity), limited to subpopulation shift (and not label shift), limited ability to verify if the conditions in theorems are satisfied in real data.


Comments to improve presentation:

Please consider contrasting the insights from this work with the fair learning under shifts literature. How does this work help in training better models under shifts?

What is the space plotted in Figure 2? Although clear from Figure 1, but useful to mention in caption.

In line 208, what is the fairness constraint the unbiased Bayes classifier will satisfy? Doesn’t it minimize the unconstrained risk? It is explicitly assumed later in Theorem 4.3 assumption 3.

Instead of linear optimization problem e.g. in line 78, please be specific that constraints are linear. Linear optimization suggests that loss function is linear which is not the case here.

Is it necessary to define risk profiles in terms of strict equality of risks across groups? Will the results extend to a more realistic case when there is small difference in risks across groups?

Line 132: “trivial random variables” -> V is an empty set.

**Main Review:**

Significance: This is my main concern. The significance of the theoretical results to evaluation and design of fair classifiers under shifts is not discussed. The condition provided in Theorem 4.3 (when to expect improvement) are hard to relate to or compute for any practical setting, thus limiting its applicability. Experiments on real world dataset also do not shed light on how the conditions can be computed. The problem of assessing future performance of the classifier under shifts is very important and the work provides a novel perspective based on geometry of risk profiles. Discussing the computable implications of the results will make them more useful for practitioners.

The work does not discuss the motivation for training fair models to correct subpopulation shift in training data. If we anticipate distribution shift, then we can use transfer learning (e.g. domain adapation, generalization) methods. It does not seem realistic to assume that modeler will not use any such methods and will rely on fairness constraints to potentially correct for the shift.

Originality: The work adds an original, geometrical perspective to studying fairness under distribution shifts. This is enabled by their focus on the space of risk profiles and the subpopulation shift condition. Related work compares with the most directly relevant papers but does not place this work in the larger fair learning under shifts literature. Some works https://cs.stanford.edu/~jure/pubs/contraction-kdd17.pdf, https://arxiv.org/abs/1806.02887, https://arxiv.org/abs/2007.06029.

Quality: The key results are technically sound. The experiments do not adequately test the results.
Not much discussion is provided on whether the conditions in Theorems 3.2, 4.3 can be tested. Both the target distribution P* and fair risk set F are unknown.
Experiments on real data too are not checking whether these conditions hold. The experiments only check for an increase in accuracy in P*. The validation of the theory is incomplete without showing that the conditions that predict this increase also hold or can reasonably be expected to hold for real data.

Clarity: The writing is clear. The figures can be explained better. Please see suggestions at the end.


Overall, I like the new geometrical perspective to reason about model performance under shifts. I would encourage authors to develop the ideas to make them more practical and to consider their implications on better evaluation or training procedures.


More detailed comments:

Replacing risk with model output in the fairness definition of demographic parity and equalised odds is assuming that the loss function used in learning is the same as that for fairness consideration. The advantage of enforcing independence with model output is that it will ensure independence with any fairness loss function. For many reasons such as ease of optimization (use of surrogate loss) or application context, the two loss functions can be different. This critical assumption should be stated.

The target for comparison in the unbiased data is assumed to be the unconstrained Bayes optimal model. I disagree with the claim that assumption 3 is implicitly assumed in many previous works, lines 229-235. The cited works are not suggesting to just use the unconstrained Bayes classifier on representative data. The classifier for unbiased data can be trained with fairness constraints too if suitable for the application context.

Label bias described in Example 4.2 differs from the subpopulation shift assumption. It is usually defined as an invariant distribution of outcome given some or all covariates, whereas the label bias and the resulting risk profile is additionally conditioning on the outcome. Please define subpopulation shifts in the Problem setup to avoid any confusion.


Questions that I would like authors to address:

Is the risk profile in 4.2 defined with respect to biased distribution P tilde?

Are the conditions for checking improvement in the theorems testable? What data does it require?

How to interpret the threshold on P* in Theorem 3.2 which is used to check whether FRM improves?

In Theorem 4.3 assumptions, which risk set or profile is used, 3.1 or 4.2?

Please clarify how the results can be used to better evaluate and design fair classifiers.

---

After the response, my concerns on the definition of subpopulation shift, and interpretation of Theorem 3.2 and Assumption 3 are addressed.

**Time Spent Reviewing:**

5

---

> ### Author Response · Authors · 2021-08-10
> **Response**
>
> We thank you for the detailed feedback and many interesting comments and suggestions. We first address the main questions in the review and then provide comments on some of your other points.
>
> ## Questions that I would like authors to address
>
> **Is the risk profile in 4.2 defined with respect to biased distribution P tilde?**
>
> In our theory, we make the subpopulation shift assumption, which implies the risk profiles with respect to the the biased training and unbiased target distributions are identical (see eq 4.4). We do note the typo in equation (4.4); the $a'$ in RHS of equation (4.4) should be $a$, i.e. the conditional loss for each subpopulations are same in expectation. In Equation (4.2), we are defining the notion of risk profile for an arbitrary distribution.
>
>
> **Are the conditions for checking improvement in the theorems testable? What data does it require?**
>
> For Thm 3.2, we need to know the fraction of minority individuals in the target domain to check whether enforcing RP will improve accuracy. For Thm 4.3, we need to have some labeled data from the target domain to check whether enforcing CRP will improve accuracy.
>
> **How to interpret the threshold on $P^\*$ in Theorem 3.2 which is used to check whether FRM improves?**
>
> Under the premise of the theorem, the minority group is underrepresented in the training data; i.e. $\widetilde{P}(A = 0) < P^*(A=0)$. Since there are only two groups, we must have $\widetilde{P}(A = 1) > P^*(A=1)$. Thus the threshold can be interpreted as the maximum level of under-representation that can be tolerated before the training and target domains are different enough that enforcing fairness starts to help, i.e. enforcing fairness will improve accuracy in the test domain. See also the paragraph immediately after Theorem 3.2. Figure 1 also provides a geometric interpretation of the threshold (see the dotted green line in the middle plot).
>
> **In Theorem 4.3 assumptions, which risk set or profile is used, 3.1 or 4.2?**
>
> We use the risk profile defined in equation (4.2). The risk profiles in (3.1) and (4.2) are related though, the first one is in the context of risk parity (where we condition only on $A$, which is typically a sensitive attribute), whereas the later one is for conditional risk parity (where we condition on both $(A, V)$ for any other attribute $V$).
>
> **Please clarify how the results can be used to better evaluate and design fair classifiers.**
>
> The main takeaway is that one needs to be careful when enforcing risk-based notions of algorithmic fairness because it can exacerbate the problem they are trying to solve. In particular, it is not a foregone conclusion that enforcing risk-based notions of algorithmic fairness will improve performance in the target domain (even under the assumptions favorable for fairness algorithms, i.e. assumption 3 in Thm 4.3). Thus practitioners should be careful and actually check that enforcing fairness is improving model performance in the target domain. For example, consider the Gender Shades [1] study which shows that the commercial gender classification algorithms are less accurate on dark-skinned individuals. A practitioner may attempt to mitigate this algorithmic bias by enforcing a RP, which leads to a fair model that sacrifices some performance on lighter-skinned individuals in exchange for improved accuracy on darker skinned individuals. The upshot from our results here is that this fair model may perform worse than the original (unfair) model in a target population with more dark-skinned individuals than in the training data. This is similar to the settings considered in our experiments, e.g. see Figure 1 (right). We will clarify this in the discussion section.
>
> ## Response to other comments and questions
>
> **The work does not discuss the motivation for training fair models to correct subpopulation shift in training data. If we anticipate distribution shift, then we can use transfer learning (e.g. domain adaption, generalization) methods. It does not seem realistic to assume that modeler will not use any such methods and will rely on fairness constraints to potentially correct for the shift.**
>
> Most existing methods for enforcing algorithmic fairness were not developed with the view that the algorithmic biases are caused by distributional shift in mind. In this paper, we study the efficacy of existing methods in mitigating algorithmic biases caused by distributional shift. We agree that a natural approach to reduce algorithmic biases caused by distributional shift is to resort to transfer learning methods, but our goal here is not to develop new methods, rather understanding the trade-off between fairness and accuracy in these settings. The important takeaway of our work is that adapting transfer learning methods to mitigate algorithmic biases is a fruitful direction for future work.
>
>
>
>
>
> **The target for comparison in the unbiased data is assumed to be the unconstrained Bayes optimal model. I disagree with the claim that assumption 3 is implicitly assumed in many previous works, lines 229-235. The cited works are not suggesting to just use the unconstrained Bayes classifier on representative data. The classifier for unbiased data can be trained with fairness constraints too if suitable for the application context.**
>
> We would like to note that in our study under RP constraint (Section 3), we don't assume the Bayes classifier for test data ($P^*$) is fair. Assumption 3 in Theorem 4.3 can be interpreted as a means to find a proper test data $P^*$ (related to a particular fairness constraint) for which performing subpopulation shift from training data is equivalent to enforcing that fairness on training data. In Section 4 we are trying to be as favourable to the common algorithmic fairness practices as possible: Bayes optimal classifier on $P^*$ is fair. Under this interpretation, Theorem 4.3 characterizes the situations when enforcing fairness on $\tilde P$ does (or doesn't) improve accuracy on $P^*$, even if the Bayes optimal test classifier is fair.
>
> **Label bias described in Example 4.2 differs from the subpopulation shift assumption. It is usually defined as an invariant distribution of outcome given some or all covariates, whereas the label bias and the resulting risk profile is additionally conditioning on the outcome. Please define subpopulation shifts in the Problem setup to avoid any confusion.**
>
> Our definition of subpopulation shift (equation (4.4)) is borrowed from the domain adaptation literature (see [2] and [3]), and Example 4.2 (label bias) provides a concrete instance of subpopulation shift. We do apologize for a typo in Equation (4.4): the $a'$ in right hand side should be $a$, i.e., we want the conditioned (on $(A, V)$) risk profiles to be equal for $P^*$ and $\tilde P$. We would also like to point out that subpopulation shift is NOT defined as ``an invariant distribution of outcome given some or all covariates''; that assumption is called covariate shift. We will clarify this in a revised version of the paper.
>
>
>
>
> **Work is limited to two fairness metrics (risk parity and conditional risk parity).**
>
>
> First, we would like to point out that Theorem 4.3 is easily generalizable for any discrete/continuous $A$ and $V$ (as defined in (4.1), the proof for the continuous case can be found in the supplementary document), although we have presented our theorem for a particular case where $V = Y$ for ease of understanding. Thus, our theory applies to many fairness constraints which fall under the setup in Equation (4.1), where V can be any discriminative attribute. However, our conditions do not cover calibration where one conditions on the model outcome $\hat Y$. We will clarify this in a revision.
>
>
>
>
> **In line 208, what is the fairness constraint the unbiased Bayes classifier will satisfy? Doesn’t it minimize the unconstrained risk? It is explicitly assumed later in Theorem 4.3 assumption 3.**
>
> Yes, as you said correctly, this corresponds to Assumption 3, where we assume that the unconstrained Bayes' classifier of the test distribution is fair. In Section 4 we assume the Bayes' classifier for $P^*$ satisfies CPR. We will clarify this in a revision.
>
>
> **Is it necessary to define risk profiles in terms of strict equality of risks across groups? Will the results extend to a more realistic case when there is small difference in risks across groups?**
>
> The *strict equality* assumption indicates that the risk profile lies on a linear subspace in appropriate dimension. If we change this to inequality, then the constraint set will become a polytope. The results and proofs probably will be different from their current incarnation, however our geometric view of the problem will be helpful for extending our theory to such generalization.
>
>
> ---
>
> Regarding the remaining points, we will incorporate the suggested minor changes in the revised version of the paper. We will also extend the discussion, limitations and related work sections based on your suggestions. Kindly please let us know if there are any other comments that you would like us to discuss.
>
> ---
> References
>
> [1] Buolamwini, Joy, and Timnit Gebru. Gender shades: Intersectional accuracy disparities in commercial gender classification, 2018.
>
> [2] Koh, Pang Wei, et al. Wilds: A benchmark of in-the-wild distribution shifts, 2021.
>
> [3] Santurkar, Shibani, Dimitris Tsipras, and Aleksander Madry. Breeds: Benchmarks for subpopulation shift, 2020.

---

> > ### Comment · Reviewer_jgBW · 2021-08-30
> > **Thanks for clarifications**
> >
> > I thank the authors for the clarifying the questions in detail. My concerns on the definition of subpopulation shift, and interpretation of Theorem 3.2 and Assumption 3 are addressed.
> >
> > The main remaining concerns are on verification of the conditions for the theorems. As authors say in the response, some labelled target data is required to check the sufficient condition that guarantees improvement. But, the improvement can be empirically verified using the same labelled target data. There may still be advantage in checking the condition when only a small amount of data is available or if the analyses can say something based on data from similar target domains. But the practical utility of the conditions is not stated clearly. I acknowledge the theoretical understanding the results provide. Regarding how the experiments are designed to satisfy the conditions (not verify them in cases where they may fail), my concern is addressed after reading the responses to Reviewer VZHD. I would encourage adding these clarifications to the updated paper.

---

> > > ### Author Response · Authors · 2021-08-31
> > > **Thank you**
> > >
> > > We thank the reviewer for helping us improve the paper. We will extend the experiments section explaining how the test data satisfy the conditions in Theorem 3.2 and add additional discussions based on your questions and suggestions in the revised version.

---

> ### Author Response · Authors · 2021-08-27
> **Discussion**
>
> Dear Reviewer jgBW,
>
> Thank you again for your comments and suggestions that have helped us to improve the paper. We hope you had a chance to take a look at our responses. Please let us know if you have additional questions or concerns. We look forward to having a fruitful discussion and kindly ask you to consider increasing the score if you find our responses adequate.

---

### Decision · Program_Chairs · 2021-09-27

**Decision:**

Accept (Poster)

**Comment:**

This paper examines the question of whether enforcing algorithmic fairness in the training domain will improve performance of the model in a target domain in which subpopulations (such as those defined by the sensitive attribute) occur in different proportions from the training domain.  The key assumption behind the results is that the risk profiles (expected loss conditional on group, or group and other discriminative attributes) are the same between the training and test domain.   The authors’ main result crisply characterizes conditions under which target domain performance improves.  Overall this is an interesting paper that studies an important problem, but it is not without its weaknesses.  Given the authors’ responses to concerns raised by reviewers, I believe the major weaknesses can be addressed through moderate revision and am thus recommending this paper for acceptance.

In revising the submission, the authors should focus on the following dimensions.
1. Clarity in exposition.  Reviewers jgBW VZHD enumerated a lengthy list of typos and confusions that I myself also share.  (Additionally, the difference between distributions, $P^* - \tilde P$ isn’t actually defined where it is first used in line 53.) While the authors note in their response that several reviewers described the work as well-written and clear, I want to emphasize that reviewers who went into depth in their comments noted typos, undefined and ill-defined notation, and confusion about theorem statements.  These issues are minor, but abundant.    The manuscript will be much improved once they are weeded out in revision.
2. Displaying figure 2 sideways is confusing.  If it is desired to keep the figure in this orientation to conserve space, at least rotate the text.
3. The work would benefit from further discussion of the settings in which the authors believe the results would likely apply.  That is, in what practical settings might the subpopulation shift assumption hold?  Would it hold in something like the motivating gender shades study example?
4. The COMPAS example is currently highly underdeveloped.  I would either expand on the discussion to explain what this example is illustrating and its implications for training risk assessment tools, or omit it altogether.
5. In the author response the authors have committed to making a number of revisions to address concerns raised by reviewers.  They are all worthwhile changes that will improve the clarity of the manuscript.